# Enhancing Treatment Effect Estimation via Active Learning: A Counterfactual Covering Perspective

**Hechuan Wen** [1]  **Tong Chen** [1]  **Mingming Gong** [2][3]  **Li Kheng Chai** [4]  **Shazia Sadiq** [1]  **Hongzhi Yin** [1]

## Abstract

Although numerous complex algorithms for treatment effect estimation have been developed in recent years, their effectiveness remains limited when handling insufficiently labeled training sets due to the high cost of labeling the post-treatment effect, e.g., the expensive tumor imaging or biopsy procedures needed to evaluate treatment effects. Therefore, it becomes essential to actively incorporate more high-quality labeled data, all while adhering to a constrained labeling budget. To enable data-efficient treatment effect estimation, we formalize the problem through rigorous theoretical analysis within the active learning context, where the derived key measures – *factual* and *counterfactual covering radii* determine the risk upper bound. To reduce the bound, we propose a greedy radius reduction algorithm, which excels under an idealized, balanced data distribution. To generalize to more realistic data distributions, we further propose FCCM, which transforms the optimization objective into the *Factual and Counterfactual Coverage Maximization* to ensure effective radius reduction during data acquisition. Furthermore, benchmarking FCCM against other baselines demonstrates its superiority across both fully synthetic and semi-synthetic datasets. Code: https://github.com/uqhwen2/FCCM.

## 1. Introduction

Understanding the causal effects of interventions is essential for making informed decisions, positioning treatment effect estimation as a fundamental tool with broad applications across diverse domains, including randomized control trials (RCTs) in medication (Pilat et al., 2015), A/B testing for business decision-making (Kohavi & Longbotham, 2015) and government policy evaluation (MacKay, 2020), etc. However, real-world scenarios often involve the trade-off between cost and return, e.g., the high cost of tumor imaging or biopsy at scale frequently limits the amount of treatment effects (labels) collected from the individuals given a particular drug, which in turn impacts the accuracy of estimation. Such a challenge highlights the need for designing data-efficient treatment effect estimation methods, which can be described as the following optimization problem:

$$\min_{\mathcal{S} \subset \mathcal{D}} \epsilon(f_{\mathcal{S}}) \quad \text{s.t.} \ |\mathcal{S}| \leq B, \tag{1}$$

where $\epsilon$ is a risk metric e.g. precision in estimation of heterogeneous effect (PEHE) (Shalit et al., 2017), $\mathcal{D}$ is the pool set containing all candidate samples, $f_{\mathcal{S}}$ is the regression model trained on subset $\mathcal{S}$, and $B$ is the labeling budget.

Depending on whether or not the treatment indicator $t$ is readily available alongside the covariates in $\mathcal{D}$, there are two branches of research related to the formulation in Eq. (1). If $t$ is not given, one is to build the dataset $\mathcal{S}$ via active experimental design (Addanki et al., 2022; Connolly et al., 2023; Ghadiri et al., 2024), i.e., a subset of $\mathcal{D}$ consisting only of covariates is selected, which is then partitioned to receive treatments, and subsequently annotated with treatment effects. Otherwise, given the pool set $\mathcal{D}$ consisting of both the covariates and the treatment indicator (Sundin et al., 2019; Qin et al., 2021; Jesson et al., 2021b; Wen et al., 2025), the algorithm only deals with selection from the pool set, then let the oracle label the selected samples. Although both aim to facilitate reliable, data-efficient predictions by selectively expanding the labeled dataset for training, studies on the latter are scarcer compared with those on the former. Despite the under-exploration of the latter setting, it has been widely encountered in real-world applications, such as collecting customer preferences from those who have already received different services, tracking patients' side effects from those who have already been administered different drugs, etc. Therefore, in this paper, we study the latter scenario where both the covariates and the treatment indicator are known in the pool set, while in the presence of high treatment effect

[1]School of EECS, The University of Queensland, Australia [2]School of Mathematics and Statistics, The University of Melbourne, Australia [3]Department of Machine Learning, Mohamed bin Zayed University of Artificial Intelligence, United Arab Emirates [4]Health and Wellbeing Queensland, Australia. Correspondence to: Hongzhi Yin <h.yin1@uq.edu.au>.

*Proceedings of the 42$^{nd}$ International Conference on Machine Learning*, Vancouver, Canada. PMLR 267, 2025. Copyright 2025 by the author(s).

annotation cost and a limited labeling budget, only a subset of them are selected for the oracle to label.

Unlike traditional optimization problems where abundant training data is assumed for treatment effect estimation (Shalit et al., 2017; Shi et al., 2019; Jesson et al., 2020; Wang et al., 2024), Eq. (1) is in-essence an active learning (AL) problem (Settles, 2009) and considers the practical issue of the scarcity of labeled training data, in conjunction with a limited labeling budget to expand the dataset for training a more generalizable model. Despite the simple formulation, solving the problem is NP-hard due to the combinatorial nature of selecting the optimal subset of data points to label (Tsang et al., 2005; Settles, 2009). From a general AL perspective, by simplifying the pre-assigned treatment as a feature variable, AL-based regression methods (Gal et al., 2017; Sener & Savarese, 2018; Ash et al., 2019) might be directly applicable to Eq. (1). However, straightforward adoption of AL for Eq. (1) is suboptimal as it omits the distribution alignment between different treatment groups during data acquisition.

Thus far, some designated AL approaches have been proposed to address the constrained regression problem in Eq. (1) with observational data. Qin et al. (2021) formalize a theoretical framework QHTE that directly extends the treatment effect estimation risk upper bound (Shalit et al., 2017) by the core-set (Tsang et al., 2005) approach; however, the derived optimizable objectives do not account for the importance of promoting distribution alignment during label acquisition. Following that, Jesson et al. (2021b) propose $\mu\rho$BALD from an information theory perspective, which reduces the distributional discrepancy between treatment groups by scaling the acquisition criterion with the inverse of counterfactual uncertainty. However, such a method relies heavily on the accuracy of the quantified uncertainty and the training of complex estimators, e.g., deep kernel learning (Wilson et al., 2016). To get the best of both worlds, Wen et al. (2025) devise a simple yet performant algorithm MACAL, which considers the reduction of distributional discrepancy while remaining model-independent during data acquisition. However, MACAL has to query the data in pairs (i.e., one each from treated/control group), which hinders its generalizability when optimality can be achieved by querying from one treatment group. Additionally, it is hard to obtain the overall risk upper bound convergence despite the convergence analysis on sub-objectives.

**Contribution.** Considering the aforementioned theoretical and practical limitations of existing methods for data-efficient treatment effect estimation, this paper presents a three-fold contribution to this area of research. 1). We establish a theoretical framework rooted in the active learning paradigm, specifically tailored to address Eq. (1) which is not directly optimizable. Unlike $\mu\rho$BALD (Jesson

et al., 2021b), the proposed theorem outlines the *model-independent* reducible quantities as the optimization alternative, i.e., *factual* and *counterfactual* covering radius. Our theorem further generalizes QHTE by additionally accounting for the distribution alignment with the *counterfactual* covering radius. 2). In contrast to MACAL (Wen et al., 2025), we propose two model-independent algorithms, both of which obtain optimality via single data point acquisition instead of pairwise queries, to minimize the covering radius-based objective: a greedy radius reduction method that works well in idealized distributions, and a greedy *factual* and *counterfactual* coverage maximization method – FCCM that allows for greater flexibility on the data distribution of the pool set; 3). We demonstrate the superiority of FCCM on real-world covariates with extensive performance comparisons and qualitative visualizations.

## 2. Preliminaries

**Treatment effect estimation.** In this paper, we estimate the treatment effect under the potential outcome framework (Imbens & Rubin, 2015). Let the covariate, treatment, and treatment outcome space be denoted as $\mathcal{X}$, $\mathcal{T}$, and $\mathcal{Y}$, respectively. We train the treatment effect estimator $f_{\mathcal{D}} : \mathcal{X} \times \mathcal{T} \rightarrow \mathcal{Y}$ based on the dataset $\mathcal{D} = \{(\mathbf{x}_i, t_i, y_i)\}_{i=1}^{N}$, where $\mathbf{x}_i$, $t_i$, $y_i$ are respectively the feature vectors, treatment assignment, and treatment outcome that correspond to the $i$-th individual. Without loss of generality, we consider binary treatments $t \in \{0, 1\}$, and use $Y^{t=1}$ and $Y^{t=0}$ to denote the potential outcomes of corresponding treatments. The ground truth individual treatment effect (ITE) for individual $\mathbf{x}$ is defined as (Shalit et al., 2017):

$$\tau(\mathbf{x}) = \mathbb{E}[Y^{t=1} - Y^{t=0} \mid \mathbf{x}]. \tag{2}$$

To evaluate the performance of the trained model $f_{\mathcal{D}}$, we adopt the expected precision in estimation of heterogeneous effect (PEHE) (Hill, 2011), which is the go-to choice for various treatment effect estimation tasks (Shalit et al., 2017; Louizos et al., 2017; Shi et al., 2019; Jesson et al., 2021b; Wang et al., 2024):

**Definition 2.1.** *The expected PEHE of the estimator $f$ with squared loss metric $\xi(\cdot)$ is defined as:*

$$\epsilon_{\text{PEHE}}(f) = \int_{\mathcal{X}} \xi(\mathbf{x}; f) p(\mathbf{x}) d\mathbf{x}, \tag{3}$$

where $\xi(\mathbf{x}; f) = (\hat{\tau}(\mathbf{x}) - \tau(\mathbf{x}))^2$, $\tau(\mathbf{x})$ is the ground truth effect defined in (2), and $\hat{\tau}(\mathbf{x}) = f(\mathbf{x}, t = 1) - f(\mathbf{x}, t = 0)$ is its estimation. The lower the PEHE value, the better the model performance.

For the identifiability of the treatment effect $\tau(\mathbf{x})$, the following assumptions from the causal inference literature are the sufficient conditions to let it hold (Shalit et al., 2017; Pearl, 2009):

**Assumption 2.2** (Consistency). Only one potential outcome is seen by each unit given the treatment $t$, i.e., $y = Y^{t=0}$ if $t = 0$ or $y = Y^{t=1}$ if $t = 1$.

**Assumption 2.3** (Strong Ignorability). The independence relation $\{Y^{t=0}, Y^{t=1}\} \perp\!\!\!\perp t \mid \mathbf{x}$ and the conditional probability $0 < p(t = 1 \mid \mathbf{x}) < 1$ hold for all $\mathbf{x}$.

**Integrating active learning.** In situations where the labels $y_i$ in $\mathcal{D}$ are unavailable, we thus introduce AL to build a labeled training set $\mathcal{S}$ out of $\mathcal{D}$. To enhance treatment effect estimators via AL, we 1). feed the oracle-labeled dataset $\mathcal{S} = \{(\mathbf{x}_i, t_i, y_i)\}_{i=1}^{k}$ for training estimator $f_{\mathcal{S}}$; 2). evaluate the performance of the model; 3). determine if training can be terminated based on performance or labeling budget; 4). if *no* to the above, select the unlabeled subset $\tilde{\mathcal{S}}^*$ from pool set $\mathcal{D} = \{(\mathbf{x}_i, t_i)\}_{i=1}^{n}$ for the oracle to label; 5). expand the fully labeled subset $\mathcal{S}$ with newly labeled data points $\tilde{\mathcal{S}}^*$ and return to step 1). This recursive procedure terminates when the desired performance is reached or the labeling budget is exhausted, and to achieve the objective in Eq. (1), the key is to identify the best strategy to construct $\mathcal{S}$.

**Related work.** In addition to the analysis in Section 1, we include a more detailed review of existing work on treatment effect estimation, AL, and data-efficient treatment effect estimation in Appendix B.

## 3. Bounds: A Counterfactual Covering Perspective

We upper-bound Eq. (1) in a general form under the AL paradigm, using a formulation similar to that defined in (Sener & Savarese, 2018) for the classification problem. Thus, the population risk $-\epsilon_{\text{PEHE}}(f_{\mathcal{S}})$ is constrained as:

$$
\begin{aligned}
&\epsilon_{\text{PEHE}}(f_{\mathcal{S}}) \\
=&\epsilon_{\text{PEHE}}(f_{\mathcal{S}}) - \frac{1}{n}\sum_{i=1}^{n}\xi(\mathbf{x}_i; f_{\mathcal{S}}) + \frac{1}{n}\sum_{i=1}^{n}\xi(\mathbf{x}_i; f_{\mathcal{S}}) - \\
&\frac{1}{|\mathcal{S}|}\sum_{j=1}^{|\mathcal{S}|}l(\mathbf{x}_j, y_j, t_j; f_{\mathcal{S}}) + \frac{1}{|\mathcal{S}|}\sum_{j=1}^{|\mathcal{S}|}l(\mathbf{x}_j, y_j, t_j; f_{\mathcal{S}}) \\
\leq& \underbrace{\left|\epsilon_{\text{PEHE}}(f_{\mathcal{S}}) - \frac{1}{n}\sum_{i=1}^{n}\xi(\mathbf{x}_i; f_{\mathcal{S}})\right|}_{\text{Generalization Error}} + \\
&\underbrace{\left|\frac{1}{n}\sum_{i=1}^{n}\xi(\mathbf{x}_i; f_{\mathcal{S}}) - \frac{1}{|\mathcal{S}|}\sum_{j=1}^{|\mathcal{S}|}l(\mathbf{x}_j, y_j, t_j; f_{\mathcal{S}})\right|}_{\text{Subset Generalization Gap } \Delta} + \\
&\underbrace{\frac{1}{|\mathcal{S}|}\sum_{j=1}^{|\mathcal{S}|}l(\mathbf{x}_j, y_j, t_j; f_{\mathcal{S}})}_{\text{Empirical Training Loss}},
\end{aligned}
\tag{4}
$$

where $\xi(\cdot; f_{\mathcal{S}})$ is in Definition 2.1 (incalculable without the counterfactual outcomes), and $l(\cdot; f_{\mathcal{S}})$ a loss function for the labeled training set $\mathcal{S}$ with observed potential outcomes.

The expected model risk $\epsilon_{\text{PEHE}}(f_{\mathcal{S}})$ at the population level with trained estimator $f_{\mathcal{S}}$ on subset $\mathcal{S}$ is controlled by three terms as shown in Eq. (4). The generalization error is bounded w.r.t. the size $n$ as in the general machine learning research (Vapnik, 1999). Commonly, the counterfactual effect for the same unit is rarely observable, e.g., a patient undergoes only one type of surgery, or a customer experiences only a single version of the software in an A/B test. Consequently, rendering the term $\xi(\mathbf{x}_i; f_{\mathcal{S}})$ non-computable. Nonetheless, the key is that we can bound the gap between the critical yet incalculable term with the training loss on $\mathcal{S}$ in the remaining two terms. By assuming zero training loss in the same fashion as (Sener & Savarese, 2018), we can explicitly formalize Eq. (1) into:

$$
\min_{\mathcal{S}\subset\mathcal{D}}\left|\frac{1}{n}\sum_{i=1}^{n}\xi(\mathbf{x}_i; f_{\mathcal{S}}) - \frac{1}{|\mathcal{S}|}\sum_{j=1}^{|\mathcal{S}|}l(\mathbf{x}_i, y_i, t_i; f_{\mathcal{S}})\right|
\tag{5}
$$
$$
\text{s.t.}\quad |\mathcal{S}| \leq B,
$$

where the subset generalization gap $\Delta$ in $\epsilon_{\text{PEHE}}(f_{\mathcal{S}})$ is maintained as the final objective.

To optimize the objective in Eq. (5) that is incalculable because of $\xi(\mathbf{x}_i; f_{\mathcal{S}})$, we instead focus on identifying reducible quantities by capping the objective with a probabilistic risk upper bound outlined in Theorem 3.4. The key reducible quantities, i.e., factual covering radius $\delta_{(t,t)}$, and the counterfactual covering radius $\delta_{(t,1-t)}$ are defined below.

**Definition 3.1.** The covering radius $\delta_{(t,t')}$ is defined as the radius of the smallest ball centered at the labeled samples from treatment group $t$, such that the union of these balls covers all samples within group $t'$, $\forall t' \in \{t, 1-t\}$, with factual covering induced by $t' = t$ and counterfactual covering induced by $t' = 1-t$.

To derive Theorem 3.4, we further assume the Lipschitz continuity and the existence of the constant $\kappa$, with the discussions on their practicability given in Appendix C.5.

**Assumption 3.2** (Lipschitz Continuity). Assume that the conditional probability density function $p^t(y|\mathbf{x})$ is $\lambda_t$-Lipschitz, the squared loss function $l$ is $\lambda_l$-Lipschitz and $l$ is further bounded by $L_l$.

**Assumption 3.3** (Constant $\kappa$). Let $\mathcal{H} = \{h|h : \mathcal{X} \rightarrow \mathbb{R}\}$ be a family of functions and $f : \mathcal{X} \times \mathcal{T} \rightarrow \mathcal{Y}$ be the hypothesis. Assume that there exists a constant $\kappa > 0$, such that $h_f(\mathbf{x}, t) := \frac{1}{\kappa}l_f(\mathbf{x}, t) \in \mathcal{H}$.

**Theorem 3.4.** *Let $\mathbf{x}$ be sampled i.i.d. $n$ times from domain $\mathcal{X}$. Under Assumption 3.2 and Assumption 3.3, with probability at least $1 - \gamma$, where $\gamma \in (0, 1)$, the subset*

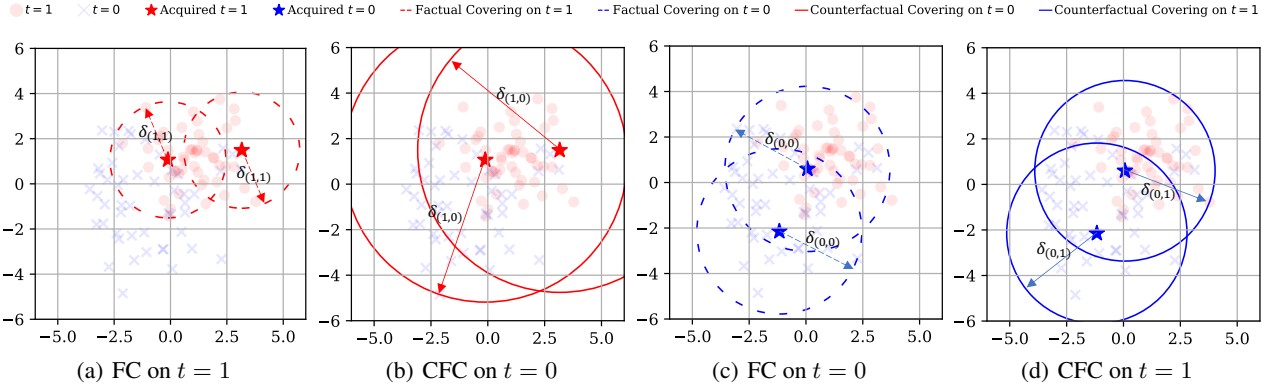

(a) FC on $t = 1$      (b) CFC on $t = 0$      (c) FC on $t = 0$      (d) CFC on $t = 1$

*Figure 1.* Visualization of the factual covering (FC) and the counterfactual covering (CFC) on the dataset by the acquired samples from each group. Note that each covering is constrained by the full coverage on the desired dataset with the minimum radius.

*generalization gap $\Delta$ is upper-bounded as:*

$$\left| \frac{1}{n} \sum_{i=1}^{n} \xi(\mathbf{x}_i; f_{\mathcal{S}}) - \frac{1}{|\mathcal{S}|} \sum_{j=1}^{|\mathcal{S}|} l(\mathbf{x}_i, y_i, t_i; f_{\mathcal{S}}) \right|$$

$$\leq \sum_{t \in \{0,1\}} \kappa_t \left( \delta_{(t,t)} + \delta_{(t,1-t)} \right) + 2\kappa_{\mathcal{H}} + \sqrt{\frac{L_l^2 \log \frac{1}{\gamma}}{2n}},$$

(6)

*where the constants $\kappa_t = 2\left(\lambda_l + \frac{1}{3}\lambda_t L_l^{\frac{3}{2}}\right)$, and $\kappa_{\mathcal{H}} = \kappa \cdot IPM_{\mathcal{H}}(p^{t=1}(\mathbf{x}), p^{t=0}(\mathbf{x}))$ with $IPM_{\mathcal{H}}(\cdot, \cdot)$ being the integral probability metric induced by $\mathcal{H}$, and $p^t(\mathbf{x})$ denotes the density distribution of treatment group $t$.*

Proof of the theorem is provided in Appendix A.1. To enhance interpretation, we visualize the factual and counterfactual covering radius in Figure 1 with four labeled samples from a random two-dimensional toy dataset. For example, Figure 1(a) shows the factual covering with radius $\delta_{(1,1)}$ such that the union of the circles centered at the two labeled $t = 1$ samples covers *all* samples from group $t = 1$. Noting that the covering radius $\delta_{(t,t')}$ decreases monotonically as the size of labeled set $\mathcal{S}$ grows under the AL paradigm, we present a further corollary to reveal the convergence of the subset generalization gap $\Delta$ given the fixed-size pool set $\mathcal{D}$.

**Corollary 3.5** (Informal). *Let $n$ be fixed, given $\delta_{(t,t')}$ decreases monotonically as $\mathcal{S}$ grows under the AL paradigm, then, with probability at least $1 - \gamma$, we have:*

$$\Delta = \mathcal{O}(\delta_{(1,1)}) + \mathcal{O}(\delta_{(1,0)}) + \mathcal{O}(\delta_{(0,0)}) + \mathcal{O}(\delta_{(0,1)}). \quad (7)$$

Qualitatively, the factual covering radius is inversely related to the sample diversity, e.g., if the two marked samples in Figure 1(a) are overlapped (low diversity), the radius $\delta_{(1,1)}$ is larger. The counterfactual covering radius is also inversely related to the distributional discrepancy, e.g., if the two marked $t = 1$ samples in Figure 1(b) are further

away from the center of the group $t = 0$, the radius $\delta_{(1,0)}$ is larger. Thus, it is evident that the objective is to minimize the covering radius as much as possible within the labeling budget, thereby reducing the risk upper bound. Intuitively, the radius reduction given limited centers is analogous to the $k$-Center problem (Wolf, 2011). However, under the treatment effect estimation setting, the derived bound introduces the counterfactual covering radius which fundamentally differs from the classical $k$-Center problem, where the centers only cover data points from the same class (i.e., treatment group in our case), as per Figure 1(a) and 1(c).

## 4. Methodologies

In this section, we propose two greedy algorithms to minimize the four covering radii in Eq. (7). The first algorithm draws direct inspiration from the Corollary 3.5 and extends the core-set solution (Tsang et al., 2005; Sener & Savarese, 2018) into the counterfactual covering perspective, while the second algorithm provides more flexibility on the data distribution and relaxes the full coverage constraint to achieve stronger radius reduction under the same labeling budget.

### 4.1. Factual and Counterfactual Radii Reduction

Denote $\mathbf{x}_i^t$, $\mathcal{D}_t$, and $\mathcal{S}_t$ as the individual covariate vector, pool set, and labeled training set for the treatment group $t$, respectively. Note that $d(\cdot, \cdot)$ is a distance metric, and $\tilde{\mathcal{S}}_t$ is a proxy collection which is explicitly explained in Appendix C.1. Motivated by the Corollary 3.5, the objective is to find the optimal subset $\mathcal{S}$ that minimizes the sum of the factual and the counterfactual covering radii as follows:

$$\min_{\mathcal{S}=\mathcal{S}_0 \cup \mathcal{S}_1, |\mathcal{S}| \leq B} \delta_{(1,1)} + \delta_{(1,0)} + \delta_{(0,0)} + \delta_{(0,1)}, \quad (8)$$

and for $t \in \{0, 1\}$, $\delta_{(t,t)} = \max_{i \in \mathcal{D}_t \setminus \mathcal{S}_t} \min_{j \in \mathcal{S}_t} d(\mathbf{x}_i^t, \mathbf{x}_j^t)$, $\delta_{(t,1-t)} = \max_{i \in \mathcal{D}_{1-t} \setminus \tilde{\mathcal{S}}_{1-t}} \min_{j \in \mathcal{S}_t} d(\mathbf{x}_i^{1-t}, \mathbf{x}_j^t)$.

However, to say the least, this minimization is as difficult

**Algorithm 1** Greedy Radius Reduction (*Sketch*)

1: **Input:** $\mathcal{D}_1, \mathcal{D}_0$; randomly initialized $\mathcal{S}_1, \mathcal{S}_0$; budget $B$; pseudo operator $\Gamma := \arg\max\min d(\cdot, \cdot)$
2: $\mathcal{S}^{\text{init}} \leftarrow \mathcal{S}_1 \cup \mathcal{S}_0, \mathcal{S} \leftarrow \mathcal{S}^{\text{init}}, \tilde{\mathcal{S}}_1 \leftarrow \varnothing, \tilde{\mathcal{S}}_0 \leftarrow \varnothing$
   #Labels in $\mathcal{S}$ is not used in querying.
3: **while** $|\mathcal{S}| < |\mathcal{S}^{\text{init}}| + B$ **do**
4:    Calculate $\delta \leftarrow \max\{\delta_{(1,1)}, \delta_{(1,0)}, \delta_{(0,0)}, \delta_{(0,1)}\}$
5:    **if** $\delta == \delta_{(1,1)}$ or $\delta == \delta_{(0,0)}$ **then**
6:       Find point $a$ to reduce the factual radius via $\Gamma$.
7:    **else**
8:       Find proxy point $a'$ to reduce the counterfactual radius via $\Gamma$, and then compute corresponded $a$.
9:    **end if**
10:    $\mathcal{S} \leftarrow \mathcal{S} \cup \{a\}$ #$a$ is not labeled.
11: **end while**
12: **Output:** $\mathcal{S}$

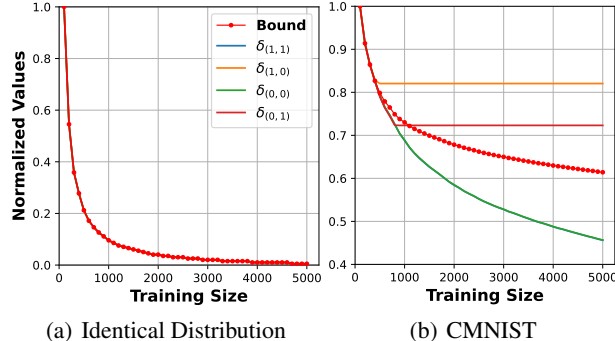

(a) Identical Distribution   (b) CMNIST

*Figure 2.* Visuals of the radius reduction and the descent of the Bound under ideal and realistic data distributions by Algorithm 1.

as solving the classic $k$-Center problem that is NP-hard (Cook et al., 1998). The minimization of the factual covering radii, i.e., $\delta_{(1,1)}$ and $\delta_{(0,0)}$, is essentially the classical $k$-Center problem, while the minimization of the counterfactual covering radii, i.e., $\delta_{(1,0)}$ and $\delta_{(0,1)}$, involves the unconventional covering of one group by the centers from the other group, e.g., covering *all* samples within group $t = 1$ by the centers from group $t = 0$ or vise versa, as visually depicted in Figure 1(b) and 1(d).

To solve Eq. (8), we provide a greedy radius reduction method in Algorithm 1 (*sketch*, with details in Appendix C.1), where the largest radius among the four radii is prioritized to be reduced. Recall Assumption 2.3 (Strong Ignorability), the counterfactual covering radii returned by our proposed algorithm can be effectively reduced under this assumption for the data distribution between the treatment groups, i.e., $0 < p(t = 1|\mathbf{x}) < 1$. Furthermore, denoting the minimal covering radius under the optimal solution as $OPT_{\delta_{(\cdot,\cdot)}}$, we give the theoretical guarantee for Algorithm 1 as follows:

**Theorem 4.1.** *Under Assumption 2.3, the sum of the covering radii returned by Algorithm 1 is upper-bounded by* $2 \times \sum_{t \in \{0,1\}} (OPT_{\delta_{(t,t)}} + OPT_{\delta_{(t,1-t)}})$.

Proof is provided in Appendix A.5. It is noted that data distribution can strongly affect the effectiveness of radius reduction by Algorithm 1. For example, if two treatment groups share identical distributions, Eq. (8) simply reduces to the conventional $k$-Center problem, for which a quick convergence of the covering radii is foreseeable. However, partially overlapped data distributions between groups can bring exponential challenges in effectively minimizing the sum of radii. In Figure 2, we illustrate the reduction of radii and their sum (denoted as Bound) in the normalized form against the size of acquired data, e.g., 1.0 represents the max value, 0.5 represents half of the max value. Under the

identical distribution scenario, in Figure 2(a), the factual and counterfactual covering radii decline synchronously (five plots fully overlap with each other) under a quick risk convergence to zero, because Eq. (8) is reduced to the simple $k$-Center problem (where Assumption 2.3 surely satisfies) which guarantees a $2 - \text{OPT}$ approximation and the greedy nature of Algorithm 1 forces the synchronous reduction. However, real-world data distribution commonly witnesses large group-wise discrepancies, e.g., in CMNIST benchmark (Jesson et al., 2021a), a significantly slower bound convergence is observed in Figure 2(b) due to the difficulty of consistently reducing the counterfactual covering radii $\delta_{(1,0)}$ and $\delta_{(0,1)}$. Note that the synchronous reduction of $\delta_{(1,1)}$ and $\delta_{(0,0)}$ is also forced by the greedy nature of Algorithm 1.

**Discussions:** Given the probability threshold $\gamma$ and fixed-size pool set $\mathcal{D}$, Theorem 3.4 provides an adjustable upper-bound which depends on the declining covering radii as $\mathcal{S}$ grows. However, the derivation of the upper bound is constrained by the full coverage on the pool set, which challenges the counterfactual covering radius reduction due to the uncontrollable real-world data distribution. In light of this, we further explore the possibility of relaxing the full coverage constraint to a more flexible condition, i.e., approximating the full coverage given the relatively smaller fixed covering radius. This underlying mentality resonates with the duality discussion of the core-set and the max coverage problem in (Yehuda et al., 2022). In the following section, we propose the factual and counterfactual coverage maximization (FCCM) solution to solve the radius reduction problem under compromised data distributions while maintaining high satisfaction with the constraint.

### 4.2. Counterfactual-integrated Coverage Maximization

Let the factual and the counterfactual covering balls, for treatment group $t$ be defined as follows:

**Definition 4.2.** Given the fixed covering radius $\delta_{(t,t')} > 0$,

**Algorithm 2** FCCM

1: **Input:** Covariate matrix $\mathbf{X} \in \mathcal{D}$, random $\mathcal{S}^{\text{init}}$; $\mathcal{S} \leftarrow \mathcal{S}^{\text{init}}$; radius $\delta_{(t,t')}, \forall t, t' \in \{0, 1\}$; weight $\alpha$; budget $B$

2: $\mathcal{E}_{(t,t)} = \{(\mathbf{x}^t, \mathbf{x}') : \mathbf{x}' \in \mathcal{A}_F^t(\mathbf{x}^t)\}$, $\mathcal{E}_{(t,1-t)} = \{(\mathbf{x}^t, \mathbf{x}') : \mathbf{x}' \in \mathcal{A}_{CF}^t(\mathbf{x}^t)\}$, $\mathcal{E}^{t=1} = \mathcal{E}_{(t,t)} \cup \mathcal{E}_{(t,1-t)}$

$$W(\mathbf{x}^t, \mathbf{x}') = \begin{cases} 1, & \text{if } \mathbf{x}' \in \mathcal{A}_F^t(\mathbf{x}^t) \\ \alpha, & \text{if } \mathbf{x}' \in \mathcal{A}_{CF}^t(\mathbf{x}^t) \end{cases},$$

Weighted $\mathcal{G} = (\mathcal{V} = \mathbf{X}, \mathcal{E} = \mathcal{E}^{t=1} \cup \mathcal{E}^{t=0}, \mathcal{W} = W)$

3: **for** $v \in \mathcal{S}^{\text{init}}$ **do**

4:     Remove the edges to the covered vertices: $\{(\mathbf{x}', \mathbf{x}^t) \in \mathcal{E}_{(t,t)}, (\mathbf{x}^t, \mathbf{x}') \in \mathcal{E}_{(t,1-t)} : (v^t, \mathbf{x}^t) \in \mathcal{E}_{(t,t)}, \forall t \in \{0, 1\})\}$

5: **end for**

6: **while** $|\mathcal{S}| \leq |\mathcal{S}^{\text{init}}| + B$ **do**

7:     $\mathcal{S} \leftarrow \mathcal{S} \cup \{v\}$, where $v$ is the vertice with the highest scaled out-degree in graph $\mathcal{G}$ (see Appendix C.2).

8:     Remove the edges to the covered vertices: $\{(\mathbf{x}', \mathbf{x}^t) \in \mathcal{E}_{(t,t)}, (\mathbf{x}^t, \mathbf{x}') \in \mathcal{E}_{(t,1-t)} : (v^t, \mathbf{x}^t) \in \mathcal{E}_{(t,t)}), \forall t \in \{0, 1\}\}$

9: **end while**

10: **Output:** $\mathcal{S}$

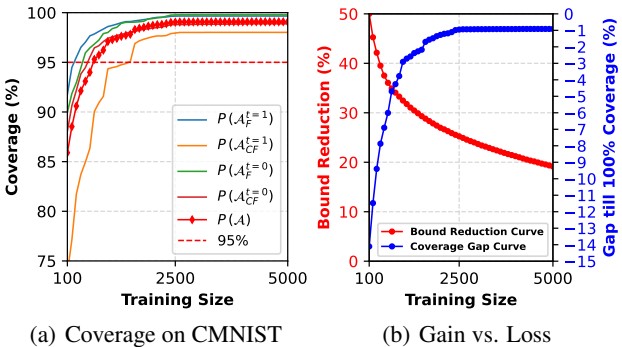

*Figure 3.* Visualization of the high coverage by Algorithm 2 on CMNIST, and reduction gain over mean coverage loss by Algorithm 2 when compared to Algorithm 1.

the covering ball $\mathcal{A}_{(t,t')}$ centered at $\mathbf{x} \in \mathcal{S}_t$ is defined as: $\mathcal{A}_{(t,t')}(\mathbf{x}) = \{\mathbf{x}' \in \mathcal{D}_{t'} : \|\mathbf{x} - \mathbf{x}'\| \leq \delta_{(t,t')}\}, \forall t' \in \{t, 1 - t\}$, with the factual covering ball induced by $t' = t$ and the counterfactual covering ball induced by $t' = 1 - t$.

Therefore, let the union of the factual covering balls be $\mathcal{A}_F^t = \bigcup_{\mathbf{x} \in \mathcal{S}_t} \mathcal{A}_{(t,t)}(\mathbf{x})$ and the factual coverage be $P(\mathcal{A}_F^t) = \frac{|\mathcal{A}_F^t|}{|\mathcal{S}_t|} \in (0, 1]$; let the union of the counterfactual covering balls be $\mathcal{A}_{CF}^t = \bigcup_{\mathbf{x} \in \mathcal{S}_t} \mathcal{A}_{(t,1-t)}(\mathbf{x})$ and the counterfactual coverage be $P(\mathcal{A}_{CF}^t) = \frac{|\mathcal{A}_{CF}^t|}{|\mathcal{S}_{1-t}|} \in [0, 1]$. Then, we further define the mean coverage $P(\mathcal{A})$ and sum of the radii $\delta_{\text{sum}}$:

$$P(\mathcal{A}) = \frac{1}{4}(P(\mathcal{A}_F^{t=1}) + P(\mathcal{A}_{CF}^{t=1}) + P(\mathcal{A}_F^{t=0}) + P(\mathcal{A}_{CF}^{t=0})),$$
$$\delta_{\text{sum}} = \delta_{(1,1)} + \delta_{(1,0)} + \delta_{(0,0)} + \delta_{(0,1)}. \quad (9)$$

With the underlying full coverage constraint, Eq. (8) can be explicitly expressed as:

$$\min_{\mathcal{S} \in \mathcal{D}} \delta_{\text{sum}} \quad \text{s.t.} \quad P(\mathcal{A}) - 1 = 0. \quad (10)$$

Noting that in Section 4.1, we discuss the dilemma of reducing $\delta_{(t,1-t)}$ due to the large discrepancy that exists in the realistic dataset. To further suppress the interested bound – $\delta_{\text{sum}}$, we transform Eq. (10) into the mean coverage maximization in Eq. (11) to maximally satisfy the equality constraint given smaller radius for the bound:

$$\max_{\mathcal{S} \in \mathcal{D}} P(\mathcal{A}). \quad (11)$$

To solve (11), we propose a greedy solution – factual and counterfactual coverage maximization (FCCM) in Algorithm 2. Specifically, FCCM constructs a weighted graph $\mathcal{G}$ with the node $V$ by the entire covariate matrix $\mathbf{X} \in \mathcal{D}$, and each node $v^t \in \mathcal{D}_t$ builds the directed edge $e(v^t, u^t)$ (with unit weight) pointing to the node $u^t$ within the ball $\mathcal{A}_{(t,t)}(v^t)$, thus constructing a directed graph $\mathcal{G}_t$. Then, graph $\mathcal{G}_t$ is further expanded by adding the weighted edges $e(v^t, u^{1-t})$ (with weight $\alpha$) for $v^t \in \mathcal{D}_t$ pointing to the node $u^{1-t}$ within the counterfactual ball $\mathcal{A}_{(t,1-t)}(v^t)$. Once $\mathcal{G}_{t=1}$ and $\mathcal{G}_{t=0}$ are both obtained, they are naturally connected to build the final weighted graph $\mathcal{G}$. Note that FCCM differs from ProbCover (Yehuda et al., 2022) which only works on a single class/group, whereas FCCM not only handles binary treatment groups but also integrates the distinctive counterfactual covering to solve Eq. (11).

Unlike Algorithm 1, which can be regarded as a *top-down* approach that optimizes the risk upper bound throughout the AL process with the equality constraint satisfied at each step, Algorithm 2 adopts a *bottom-up* approach, working on satisfying the equality constraint under a fixed bound. To further give Algorithm 2 a theoretical guarantee in Theorem 4.4 (with proof provided in Appendix A.6), we assume that the full factual and counterfactual coverage are possible by the returned labeled set $\mathcal{S}$ as follows:

**Assumption 4.3.** Given the fixed covering radius $\delta_{(t,t)}$ and $\delta_{(t,1-t)}$, there exists the optimal solution $\mathcal{S}_t^*, \mathcal{S}_t^* \subset \mathcal{S}^*$ for treatment group $t$, $\forall t \in \{0, 1\}$, such that $\mathcal{A}_F^t \bigcup \mathcal{A}_{CF}^t = \mathcal{D}$.

**Theorem 4.4.** *Under Assumption 4.3, Algorithm 2 is a $(1 - \frac{1}{e})$ – approximation for the full coverage constraint on the equally weighted graph and unscaled out-degree.*

**Exploratory analysis**: Following our discussions on Algorithm 1, we conduct the exploratory analysis on the CM-NIST benchmark (Jesson et al., 2021a) in Figure 3(a) to demonstrate the high coverage efficiency by Algorithm 2 on the real-world covariates. Further experiment on the per-

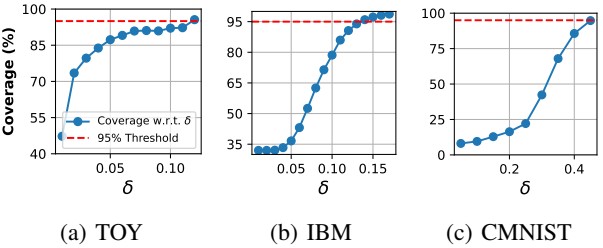

(a) TOY      (b) IBM      (c) CMNIST

*Figure 4.* Estimating a smaller range for the covering radius $\delta$ around the 95% coverage threshold by Algorithm 2.

formance compared to Algorithm 1 is accessible in Figure 3(b), where both the high reduction of the sum of the radii and the low compromise of the coverage are observed, e.g., maximally 25% reduction (red line) with 1% mean coverage loss (blue line) around the training size of 2500.

### 4.3. Approximating the Covering Radius

The hyperparameters, i.e., $\delta_{(t,t)}$ and $\delta_{(t,1-t)}$ are crucial for the success of obtaining a relatively lower upper bound without heavily intruding into the full coverage constraint. For simplicity, we set the same size for all four radii and denote them as $\delta$ uniformly, in Figure 4, we experiment different values of $\delta$ (normalized w.r.t. the max distance between points), and calculate the converged coverage on the growing training set given the pre-set covering radius. Thus, we estimate a narrower range for the relatively small radius with an achievable high mean coverage around the 95% threshold, to reduce the search space for further hyperparameter tuning.

**Uniform $\delta$:** The rationale to use uniform $\delta$ is to avoid making the search space prohibitively large – on the order of $\mathcal{O}(m^4)$ if each radius has $m$ candidate settings. The key insight to reduce the search complexity is that, by the definition of the mean coverage $P(\mathcal{A})$ in Eq. (9), Definition 3.1, and Definition 4.2, each radius is independent and each sub-term of $P(\mathcal{A})$ increases monotonically with its corresponding radius, which leads to the mean coverage $P(\mathcal{A})$ increases monotonically. Thus, by the independence and monotonicity, the search for four radii stays in the same direction, making the initial search space $\mathcal{O}(m)$ to identify the smallest radius for satisfying the 95% mean coverage threshold.

**Different $\delta$:** Though the shared value among four radii allows for effective and efficient initial hyperparameter tuning, it could potentially fail if the distribution discrepancy between the two treatment groups is very large. This is because the counterfactual covering radius can be far larger than the factual covering radius, i.e., $\delta_{(t,1-t)} \gg \delta_{(t,t)}$, to let the counterfactual coverage $P(\mathcal{A}_{CF}^t)$ to get close to full, lowering the utility of the uniform radius value. As such,

*Table 1.* Summary of the Acquisition Setup and Testing

| Dataset | Start | Length | Steps | Pool | Val | Test |
|---|---|---|---|---|---|---|
| TOY | ALL* | 1 | 50 | 7200 | 2880 | 1600 |
| IBM | ALL* | 50 | 50 | 2891 | 3180 | 6250 |
| CMNIST | ALL* | 50 | 50 | 16706 | 10500 | 18000 |

if the distribution discrepancy between treatment groups is large, it is then viable to identify a different value for each covering radius to maintain the high coverage under the linear complexity due to the independence and monotonicity.

## 5. Experiments

**Datasets: Toy** – a simulated 2-dimensional toy dataset based on 16,000 randomly generated samples. **IBM** (Shimoni et al., 2018) – a high-dimensional tabular dataset based on the publicly available Linked Births and Infant Deaths Database. Each simulation contains 25,000 samples with 177 real-world covariates randomly selected from a cohort of 100,000 individuals; **CMNIST** (Jesson et al., 2021a) – This dataset contains 60,000 image samples (10 classes) of size 28×28, which are adapted from MINIST (LeCun, 1998) benchmark. Further simulation details are deferred to the Appendix C.3.

**Metric:** The PEHE defined in Definition 2.1 with the squared root empirical expression: $\sqrt{\epsilon_{\text{PEHE}}} = \sqrt{\Sigma_{i=1}^N ((y_i^{t=1} - y_i^{t=0}) - \tau_i)^2/N}$, is used for measuring the risk of the estimator at the individual level.

**Baselines:** We compare FCCM to two groups of models, namely, the general AL model from the broader research field: BADGE (Ash et al., 2019), BAIT (Ash et al., 2021), and LCMD (Holzmüller et al., 2023). And the designated model for treatment effect estimation with AL: QHTE (Qin et al., 2021), Causal-Bald (Jesson et al., 2021b) (with variants $\mu$BALD, $\rho$BALD, and $\mu\rho$BALD), and MACAL (Wen et al., 2025) are the designated algorithms proposed to deal with the treatment effect estimation with AL.

**Estimators:** The same setup as described in (Jesson et al., 2021b; Wen et al., 2025) is adopted here by utilizing the following two open-source estimators: **DUE-DNN** (Van Amersfoort et al., 2021) for tabular data, and **DUE-CNN** (Van Amersfoort et al., 2021) for image data. Model training details can be found in Appendix C.6.

**Evaluation scheme:** The details of the data acquisition setup is summarized in Table 1, where we initialize the training set $\mathcal{S}$ with the entire labeled samples (denoted as ALL*) from group $t = 0$ and start acquisition only on the sample from $t = 1$, which simulates scenarios with a significant number of missing counterfactual samples. Then, a fixed step length is enforced at each acquisition step with fifty data acquisition steps. Note that for Toy dataset we

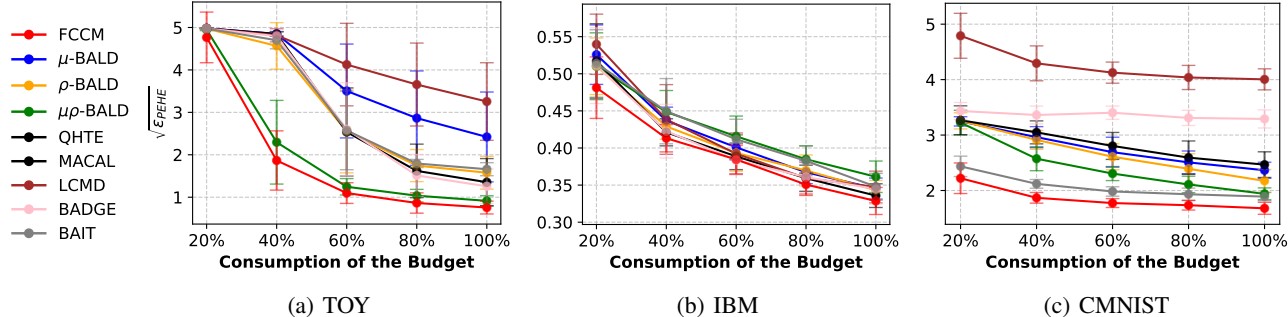

(a) TOY        (b) IBM        (c) CMNIST

*Figure 5.* All plots are the mean values averaged from 10 simulations associated with the standard deviation as the error bar. Note that all models at 0% exhibit the same performance given the fixed estimators and are thus neglected. The performance under 2% granularity is presented in Appendix C.7.

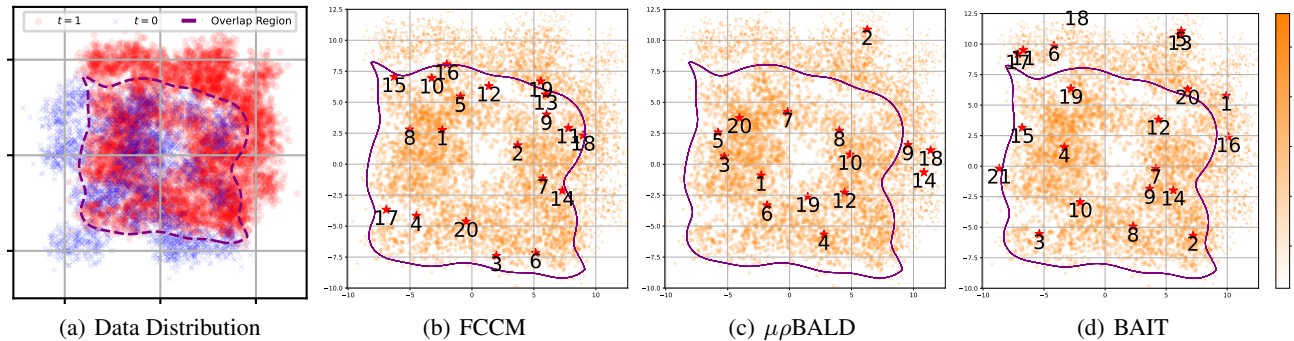

(a) Data Distribution     (b) FCCM     (c) $\mu\rho$BALD     (d) BAIT

*Figure 6.* Visualizations of the toy dataset distribution, and the actual acquisition of the data by FCCM, $\mu\rho$BALD, and BAIT. The size of the data point and color from (b) to (c) is adjusted to its associated density, with deeper color representing higher density and vice versa.

do single data acquisition to show fine-granularity results, and batch-mode acquisition for the other two datasets. Each evaluation is done by the estimator trained from the last best checkpoint without completely re-train from scratch.

### 5.1. Risk Evaluations

In Figure 5, it is observed that our proposed method is generally served as the risk lower bound in all three datasets. Its outstanding performance empirically proves the superiority of our method which considers the joint coverage on the factual and counterfactual data throughout the process of the querying. This acquisition scheme leads to a lower estimation risk that can be further explained qualitatively with the two underlying properties: i). querying from a high-density region; ii). considering the satisfactions of the positivity assumption alongside. The first property leverages the generalizability of the trained estimator to the acquired samples' neighborhoods, where a high-density neighborhood accounts for higher loss. The second property considers the pivotal overlapping assumption for treatment effect estimation, for which robust estimation toward an individual can be derived by pairing such an individual's factual or counterfactual part from the overlapping region. Thus, our proposed method outperforms the other baselines

by prioritizing the data acquisition toward the overlapping region with high density, while others cannot do both essentially.

Given the above qualitative analysis, it is explainable that the other method designed for data-efficient treatment effect estimation (DTEE), e.g., $\mu\rho$BALD, underperforms our method. For example, although $\mu\rho$BALD bias the data acquisition toward the overlapping region, it does not further embed the property to query from the high-density region, and thus accounts for less risk. The same mentality applies to the other DTEE methods, e.g., MACAL, QHTE, etc. Also, it is observed that the baselines from the general AL field, e.g., LCMD and BAIT, underperform our method by not directly considering the data acquisition toward the overlapping region, which is pivotal for treatment effect estimation. However, it is interestingly observed that BAIT can outperform many DTEE baselines on CMNIST, but it underperforms on TOY and IBM. Furthermore, the current DTEE baselines cannot consistently outperform the general AL methods across various datasets.

### 5.2. Acquisition Visualization

The original distribution of the two-dimensional toy dataset is given in Figure 6(a) for reference, where the overlapping

region is plotted in a dashed purple line by kernel density estimation (Scott, 2015). From Figure 6(c) to 6(d), the density of each data point is calculated in by Line 2 in Algorithm 2, the higher the out-degree of each node, the denser it is as visually observed in the overlapped region. Under the fine granularity of querying one sample at a time, it's observed that in Figure 6(b) FCCM consistently delivers the high priority to query from the overlapping region with high density, while $\mu\rho$BALD is witnessed to have more queries fall outside the overlapping region and less priority on query from the high-density region (query No.2 is fall at the edge of the distribution, where sparse and no overlapping is observed) as shown in Figure 6(c). Also, BAIT's acquisition spreads out the entire data on group $t = 1$ without considering the presence of the counterfactual samples and thus queries the least from the desired region as shown in Figure 6(d). Their associated performance by 20 acquired samples (40% consumption of budget) also seen with a significant gap as illustrated in Figure 5(a).

### 5.3. Ablation and Sensitivity Study

*Table 2.* Ablation Study of the Counterfactual Covering Radii

| Dataset | Method | Consumption of the Total Budget | | | | |
|---|---|---|---|---|---|---|
| | | 1/5 | 2/5 | 3/5 | 4/5 | 5/5 |
| TOY | FCCM- | 4.7680 | 2.2496 | 1.3372 | 1.0545 | 0.9024 |
| | FCCM | 4.7664 | 1.8655 | 1.0978 | 0.8637 | 0.7565 |
| | Gain | +0% | +17% | +18% | +18% | +16% |
| IBM | FCCM- | 0.4745 | 0.4088 | 0.3797 | 0.3512 | 0.3291 |
| | FCCM | 0.4813 | 0.4132 | 0.3845 | 0.3507 | 0.3286 |
| | Gain | -1% | -1% | -1% | +0% | +0% |
| CMNIST | FCCM- | 2.9250 | 2.6627 | 2.4652 | 2.3105 | 2.2073 |
| | FCCM | 2.2207 | 1.8681 | 1.7735 | 1.7344 | 1.6790 |
| | Gain | +24% | +30% | +28% | +25% | +24% |

The ablation study is conducted to study the effect of maximizing the counterfactual coverage, e.g., $P(\mathcal{A}_{CF}^{t=1})$. Essentially, two models are compared, our proposed method FCCM, and the FCCM$^-$ (by setting counterfactual covering radii $\delta_{(1,0)} = 0$ and $\delta_{(0,1)} = 0$, and this action tailors the ProbCover (Yehuda et al., 2022) to align with the context for binary-class AL). In Table 2, we capture five stages of the total query steps, and calculate the performance gain by $\frac{\sqrt{\epsilon_{\text{PEHE}}_{\text{FCCM}-}} - \sqrt{\epsilon_{\text{PEHE}}_{\text{FCCM}}}}{\sqrt{\epsilon_{\text{PEHE}}_{\text{FCCM}-}}} \times 100\%$. It is noted that the performance gain on Toy and CMNIST datasets are significant, however, with neck-to-neck performance observed on the IBM dataset. This phenomenon is explainable since the treated and control distributions on IBM are heavily overlapped such that the high-density region on the treated sample ($t = 1$) is in fact the high-density region on the counterfactual side ($t = 0$). We provide the visualization of the IBM and CMNIST in Appendix C.4 for further discussions

of the underlying rationale. Additionally, the sensitivity analysis of $\alpha$ and the covering radii $\delta_{(1,1)}$ and $\delta_{(1,0)}$ is presented in Appendix C.8. There, we generally observe that a non-zero weight $\alpha$ is influential for dataset with less overlap, and that a larger covering radius can lead to greater risk reduction in the early stage, albeit at the cost of reduced performance in the later stage.

## 6. Conclusion

We formalize the data-efficient treatment effect estimation problem under a solid theoretical framework, where the convergence of the risk upper bound is governed by the reduction of the derived factual and counterfactual covering radii. To reduce the bound, we propose a greedy radius reduction algorithm, which is $2-$OPT-like under an idealized data distribution assumption. To generalize to more realistic data distributions for higher radius reduction, we further propose FCCM, which transforms the optimization objective into the factual and counterfactual coverage maximization with a $(1 - \frac{1}{e})$–approximation to the full coverage constraint. Also, benchmarking with other baselines further proves the superiority of FCCM on solving the data-efficient treatment effect estimation problem.

**Limitation:** FCCM is designed to better handle the partially overlapped data for a quicker bound reduction while maintaining high coverage (Figure 3(b)). As such, in scenarios where the two treatment groups have non-overlapping regions in the raw feature space (e.g., biased treatment assignment), the data acquisition of FCCM will be challenged as there are no overlapping, counterfactual pairs to be identified. One possible remedy is to operate FCCM in the latent space where the inter-group distributions are aligned by methods like (Shalit et al., 2017; Zhang et al., 2020; Wang et al., 2024), but it also offsets the model-independent advantage of FCCM.

## Acknowledgements

The Australian Research Council partially supports this work under the streams of Future Fellowship (Grant No. FT210100624), Discovery Early Career Researcher Award (Grant No. DE230101033 and DE210101624), Discovery Project (Grant No. DP240101108, DP240101814 and DP240102088), Industrial Transformation Training Centre (Grant No. IC200100022), and Linkage Project (Grant No. LP230200892 and LP240200546). Additional support from Health and Wellbeing Queensland is also gratefully acknowledged. We also thank the reviewers and area chair for their insightful suggestions during the rebuttal and discussion phase.

## Impact Statement

This paper presents work whose goal is to advance the field of Causal Inference. Our work entails many potential societal benefits, including enhanced accuracy in treatment estimation for new drugs and improved robustness in causal inference models under data sparsity.

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

# A. Theory

## A.1. Proof of Theorem 3.4

**Assumption 3.2** (Lipschitz Continuity). *Assume that the conditional probability density function $p^t(y|\mathbf{x})$ is $\lambda_t$-Lipschitz, the squared loss function $l$ is $\lambda_l$-Lipschitz and $l$ is further bounded by $L_l$.*

**Assumption 3.3** (Constant $\kappa$). *Let $\mathcal{H} = \{h|h : \mathcal{X} \to \mathbb{R}\}$ be a family of functions and $f : \mathcal{X} \times \mathcal{T} \to \mathcal{Y}$ be the hypothesis. Assume that there exists a constant $\kappa > 0$, such that $h_f(\mathbf{x}, t) := \frac{1}{\kappa} l_f(\mathbf{x}, t) \in \mathcal{H}$.*

**Theorem 3.4.** *Let $\mathbf{x}$ be sampled i.i.d. $n$ times from domain $\mathcal{X}$. Under Assumption 3.2 and Assumption 3.3, with probability at least $1 - \gamma$, where $\gamma \in (0, 1)$, the subset generalization gap $\Delta$ is upper-bounded as:*

$$
\left| \frac{1}{n} \sum_{i=1}^{n} \xi(\mathbf{x}_i; f_{\mathcal{S}}) - \frac{1}{|\mathcal{S}|} \sum_{j=1}^{|\mathcal{S}|} l(\mathbf{x}_i, y_i, t_i; f_{\mathcal{S}}) \right| \tag{12}
$$
$$
\leq \sum_{t \in \{0,1\}} \kappa_t \left( \delta_{(t,t)} + \delta_{(t,1-t)} \right) + 2\,\kappa_{\mathcal{H}} + \sqrt{\frac{L_l^2 \log \frac{1}{\gamma}}{2n}},
$$

*where the constants $\kappa_t = 2\left(\lambda_l + \frac{1}{3}\lambda_t L_l^{\frac{3}{2}}\right)$, and $\kappa_{\mathcal{H}} = \kappa \cdot IPM_{\mathcal{H}}(p^{t=1}(\mathbf{x}), p^{t=0}(\mathbf{x}))$ with $IPM_{\mathcal{H}}(\cdot, \cdot)$ denotes the integral probability metric induced by $\mathcal{H}$ and $p^t(\mathbf{x})$ denotes the density distribution of treatment group $t$.*

*Proof of Theorem 3.4.* The proof is done in three main steps: Firstly, we bound the expected value of the interested term $\frac{1}{n} \sum_{i=1}^{n} \xi(\mathbf{x}_i; f_{\mathcal{S}})$ with factual and counterfactual loss, namely, $\epsilon_F$ and $\epsilon_{CF}$ over the domain of the pool set. Secondly, we constrain the $\epsilon_F$ with the factual covering radius $\delta_{(t,t)}$ and the $\epsilon_{CF}$ with the counterfactual covering radius $\delta_{(t,1-t)}$. Lastly, we conclude the probabilistic bound by Hoeffding's inequality.

$$
\mathbb{E}\left[ \frac{1}{n} \sum_{i=1}^{n} \xi(\mathbf{x}_i; f_{\mathcal{S}}) \right] \leq 2\left(\epsilon_F + \epsilon_{CF}\right) \tag{13a}
$$

$$
\leq \sum_{t \in \{0,1\}} 2\,\mu_t \left( \epsilon_{\mathcal{S}_t} + \delta_{(t,t)}(\lambda_l + \frac{1}{3}\lambda_t L_l^{\frac{3}{2}}) \right) + \sum_{t \in \{0,1\}} 2\,\mu_{1-t} \left( \epsilon_{\mathcal{S}_t} + \delta_{(t,1-t)}(\lambda_l + \frac{1}{3}\lambda_t L_l^{\frac{3}{2}}) \right) + 2\,\kappa_{\mathcal{H}} \tag{13b}
$$

$$
\leq \sum_{t \in \{0,1\}} 2\left(\lambda_l + \frac{1}{3}\lambda_t L_l^{\frac{3}{2}}\right) \left( \delta_{(t,t)} + \delta_{(t,1-t)} \right) + 2\,\kappa_{\mathcal{H}}. \tag{13c}
$$

The inequality (13a) is by Lemma A.2, the inequality (13b) is by Lemma A.3 and A.5, the equality in (13c) is by the zero training loss assumption and the fact that $\mu_t \leq 1, \forall t \in \{0, 1\}$. With Hoeffding's inequality, we have:

$$
\mathbb{P}\left( \frac{1}{n} \sum_{i=1}^{n} \xi(\mathbf{x}_i; f_{\mathcal{S}}) - \mathbb{E}\left[ \frac{1}{n} \sum_{i=1}^{n} \xi(\mathbf{x}_i; f_{\mathcal{S}}) \right] \geq \epsilon \right) = \exp\left(-\frac{2n\epsilon^2}{L_l^2}\right). \tag{13d}
$$

By setting $\gamma = \exp\left(-\frac{2n\epsilon^2}{L_l^2}\right)$, we solve for the bounding gap $\epsilon = \sqrt{\frac{L_l^2 \log \frac{1}{\gamma}}{2n}}$. Thus, we can derive that with probability at least $1 - \gamma$, the following inequality holds:

$$
\frac{1}{n} \sum_{i=1}^{n} \xi(\mathbf{x}_i; f_{\mathcal{S}}) \leq \mathbb{E}\left[ \frac{1}{n} \sum_{i=1}^{n} \xi(\mathbf{x}_i; f_{\mathcal{S}}) \right] + \sqrt{\frac{L_l^2 \log \frac{1}{\gamma}}{2n}}. \tag{13e}
$$

To conclude, we have:

$$\left| \frac{1}{n} \sum_{i=1}^{n} \xi(\mathbf{x}_i; f_{\mathcal{S}}) - \frac{1}{|\mathcal{S}|} \sum_{j=1}^{|\mathcal{S}|} l(\mathbf{x}_i, y_i, t_i; f_{\mathcal{S}}) \right| \tag{13f}$$

$$= \frac{1}{n} \sum_{i=1}^{n} \xi(\mathbf{x}_i; f_{\mathcal{S}}) \tag{13g}$$

$$\leq \mathbb{E}\left[ \frac{1}{n} \sum_{i=1}^{n} \xi(\mathbf{x}_i; f_{\mathcal{S}}) \right] + \sqrt{\frac{L_l^2 \log \frac{1}{\gamma}}{2n}} \tag{13h}$$

$$\leq \sum_{t \in \{0,1\}} 2\left(\lambda_l + \frac{1}{3}\lambda_t L_l^{\frac{3}{2}}\right)\left(\delta_{(t,t)} + \delta_{(t,1-t)}\right) + 2\,\kappa_{\mathcal{H}} + \sqrt{\frac{L_l^2 \log \frac{1}{\gamma}}{2n}}. \tag{13i}$$

The equality (13g) is by the zero training loss assumption, the inequality (13h) is by incorporating (13e), the final inequality (13i) is concluded by incorporating (13c).

**Discussion on the bound**: Given the i.i.d. sampled pool set from the domain $\mathcal{X}$, it is noted that in the final probabilistic bound, if the sampled two distributions were identical and we selected the entire pool set for training, simply by definition, we have both the factual and counterfactual covering radius be completely zero, and also the distributional discrepancy term $\kappa_{\mathcal{H}}$ counted by IPM be completed zero, leaving the tightness of the risk upper bound solely depends on the size of the pool set $n$ and for the given probability threshold $\gamma$.

$\square$

## A.2. Proof of Lemma A.2

**Definition A.1.** Given the loss metric $l$, the expected factual loss $\epsilon_F$ and counterfactual loss $\epsilon_{CF}$ are defined in a manner consistent with (Shalit et al., 2017) as follows:

$$\begin{aligned} \epsilon_F &= \int_{\mathcal{X} \times \mathcal{T}} l(\mathbf{x}, t) p(\mathbf{x}, t) d\mathbf{x} dt, \\ \epsilon_{CF} &= \int_{\mathcal{X} \times \mathcal{T}} l(\mathbf{x}, t) p(\mathbf{x}, 1-t) d\mathbf{x} dt. \end{aligned} \tag{14}$$

**Lemma A.2.** *Let $\mathbf{x}$ to be sampled i.i.d. $n$ times from the domain $\mathcal{X}$. With the two-headed trained model $f_{\mathcal{S}} = \{\hat{f}^{t=1}, \hat{f}^{t=0}\}$ on the selected subset $\mathcal{S}$, where $\hat{f}^{t=1}, \hat{f}^{t=0} : \mathcal{X} \to \mathcal{Y}$ are the estimators for the treatment effect $y^t$ respectively. The expected value of $\frac{1}{n} \sum_{i=1}^{n} \xi(\mathbf{x}_i; f_{\mathcal{S}})$ is upper-bounded as follows:*

$$\mathbb{E}\left[ \frac{1}{n} \sum_{i=1}^{n} \xi(\mathbf{x}_i; f_{\mathcal{S}}) \right] \leq 2(\epsilon_F + \epsilon_{CF}). \tag{15}$$

*Proof of Lemma A.2.* Note that the overall estimator is denoted as $f_S$, which is a two-headed model with $\hat{f}^t$ representing the estimation for the treatment effect $y^t$ under treatment assignment $t$. With a slight abuse of notation, we write the expected loss over $\mathbf{x}$ where $y$ is implicitly assumed to be a function of $\mathbf{x}$, i.e., each $\mathbf{x}$ has an associated label $y$, as is typical in

supervised learning setting:

$$\mathbb{E}\left[\frac{1}{n}\sum_{i=1}^{n}\xi(\mathbf{x}_i; f_{\mathcal{S}})\right] = \frac{1}{n}\sum_{i=1}^{n}\mathbb{E}\left[\xi(\mathbf{x}_i; f_{\mathcal{S}})\right] \tag{16a}$$

$$= \mathbb{E}_{\mathbf{x}\sim\mathcal{X}}\left[\xi(\mathbf{x}; f_{\mathcal{S}})\right] \tag{16b}$$

$$= \int_{\mathcal{X}}(\tau(\mathbf{x}) - \hat{\tau}(\mathbf{x}))^2 p(\mathbf{x})d\mathbf{x} \tag{16c}$$

$$= \int_{\mathcal{X}}((y^{t=1} - y^{t=0}) - (\hat{f}^{t=1}(\mathbf{x}) - \hat{f}^{t=0}(\mathbf{x})))^2 p(\mathbf{x})d\mathbf{x} \tag{16d}$$

$$= \int_{\mathcal{X}}\underbrace{((y^{t=1} - \hat{f}^{t=1}(\mathbf{x})) - (y^{t=0} - \hat{f}^{t=0}(\mathbf{x})))^2}_{\text{Swap } y^{t=0} \text{ and } \hat{f}^{t=1}(\mathbf{x})} p(\mathbf{x})d\mathbf{x} \tag{16e}$$

$$\leq 2\int_{\mathcal{X}}((y^{t=1} - \hat{f}^{t=1}(\mathbf{x}))^2 + (y^{t=0} - \hat{f}^{t=0}(\mathbf{x}))^2)p(\mathbf{x})d\mathbf{x} \tag{16f}$$

$$= \underbrace{2\int_{\mathcal{X}}(y^{t=1} - \hat{f}^{t=1}(\mathbf{x}))^2 p(\mathbf{x}, t=1)d\mathbf{x} + 2\int_{\mathcal{X}}(y^{t=1} - \hat{f}^{t=1}(\mathbf{x}))^2 p(\mathbf{x}, t=0)d\mathbf{x} +}_{\text{Apply } p(\mathbf{x}) = \int p(x,t)dt = p(\mathbf{x}, t=1) + p(\mathbf{x}, t=0)} \tag{16g}$$

$$2\int_{\mathcal{X}}(y^{t=0} - \hat{f}^{t=0}(\mathbf{x}))^2 p(\mathbf{x}, t=0)d\mathbf{x} + 2\int_{\mathcal{X}}(y^{t=0} - \hat{f}^{t=0}(\mathbf{x}))^2 p(\mathbf{x}, t=1)d\mathbf{x} \tag{16h}$$

$$= \underbrace{2\int_{\mathcal{X}}(y^{t=1} - \hat{f}^{t=1}(\mathbf{x}))^2 p(\mathbf{x}, t=1)d\mathbf{x} + 2\int_{\mathcal{X}}(y^{t=0} - \hat{f}^{t=0}(\mathbf{x}))^2 p(\mathbf{x}, t=0)d\mathbf{x} +}_{\text{Re-arrange and this term equals } 2\epsilon_F \text{ by definition}} \tag{16i}$$

$$\underbrace{2\int_{\mathcal{X}}(y^{t=1} - \hat{f}^{t=1}(\mathbf{x}))^2 p(\mathbf{x}, t=0)d\mathbf{x} + 2\int_{\mathcal{X}}(y^{t=0} - \hat{f}^{t=0}(\mathbf{x}))^2 p(\mathbf{x}, t=1)d\mathbf{x}}_{\text{Re-arrange and this term equals } 2\epsilon_{CF} \text{ by definition}} \tag{16j}$$

$$= 2(\epsilon_F + \epsilon_{CF}). \tag{16k}$$

$\square$

**Discussion**: Lemma A.2 indicates an interesting decomposition of the expected loss into factual and counterfactual errors, which is a prelude to our final probabilistic bound under AL paradigm. It is worth mentioning that a similar intermediate conclusion appears to align with observations in (Shalit et al., 2017).

### A.3. Proof of Lemma A.3

**Lemma A.3.** *Denote the expected loss on subset $S_t$ as $\epsilon_{S_t}$, the factual covering radius as $\delta_{(t,t)}$, let the constant $\mu_t = p(t)$ be the marginal probability. Assume that the conditional probability density function $p^t(y|\mathbf{x})$ is $\lambda_t$-Lipschitz, the squared loss function $l$ is $\lambda_l$-Lipschitz and $l$ is further bounded by $L_l$, the expected factual loss is bounded as follows:*

$$\epsilon_F \leq \sum_{t\in\{0,1\}}\mu_t\left(\epsilon_{S_t} + \delta_{(t,t)}(\lambda_l + \frac{1}{3}\lambda_t L_l^{\frac{3}{2}})\right). \tag{17}$$

*Proof of Lemma A.3.* We start with the Tower Law for the key of the proof, let the estimation be $\hat{y} = \hat{f}(\mathbf{x})$ given the fixed treatment $t$, we have the expected loss to be decomposed in the general form:

$$\mathbb{E}_{\mathcal{X}}\left[(y - \hat{f}(\mathbf{x}))^2\right] = \mathbb{E}_{\mathcal{X}}\left[\mathbb{E}_{\mathcal{Y}}\left[(y - \hat{f}(\mathbf{x}))^2 \mid \mathbf{x}\right]\right] \tag{18a}$$

$$= \mathbb{E}_{\mathcal{X}}\left[\int_{\mathcal{Y}}(y - \hat{f}(\mathbf{x}))^2 p(y \mid \mathbf{x})dy\right]. \tag{18b}$$

Denote $p^t(y|\mathbf{x}) = p(y|\mathbf{x}, t)$ and $p^t(\mathbf{x}) = p(\mathbf{x}|t)$, we apply the Tower Law conclusion in Eq. (18b) by Tow Law and further bound the expected factual error $\epsilon_F^t$ for the treatment group $t$:

$$\epsilon_F^t = \mathbb{E}_{\mathcal{X}^t}\left[(y - \hat{f}^t(\mathbf{x}))^2\right] \tag{19a}$$

$$= \mathbb{E}_{\mathcal{X}^t}\left[\mathbb{E}_{\mathcal{Y}^t}\left[(y - \hat{f}^t(\mathbf{x}))^2 \mid \mathbf{x}\right]\right] \tag{19b}$$

$$= \mathbb{E}_{\mathcal{X}^t}\left[\int_{\mathcal{Y}} l_y^t p^t(y|\mathbf{x})dy\right] \tag{19c}$$

$$= \mathbb{E}_{\mathcal{X}^t}\left[\int_{\mathcal{Y}} l_y^t(p^t(y|\mathbf{x}) \underbrace{-p^t(y|\mathbf{x}') + p^t(y|\mathbf{x}')}_{\text{Add up to Zero}})dy\right] \tag{19d}$$

$$= \mathbb{E}_{\mathcal{X}^t}\left[\int_{\mathcal{Y}} l_y^t p^t(y|\mathbf{x}')dy + \int_{\mathcal{Y}} l_y^t(p^t(y|\mathbf{x}) - p^t(y|\mathbf{x}'))dy\right]. \tag{19e}$$

We decompose the first term within the expectation in (19e) into the followings, by the selected $\mathbf{x}' \in \mathcal{S}_t$ that covers the $\mathbf{x}$ from group $t$ within the factual radius $\delta_{(t,t)}$, we bound the term as:

$$\int_{\mathcal{Y}} l_y^t p^t(y|\mathbf{x}')dy = \int_{\mathcal{Y}} (l_y^t - l_{y'}^t + l_{y'}^t)p^t(y|\mathbf{x}')dy \tag{20a}$$

$$= \int_{\mathcal{Y}} (l_y^t - l_{y'}^t)p^t(y|\mathbf{x}')dy + \int_{\mathcal{Y}} l_{y'}^t\, p^t(y|\mathbf{x}')dy \tag{20b}$$

$$\leq \delta_{(t,t)}\lambda_l + \int_{\mathcal{Y}} l_{y'}^t\, p^t(y|\mathbf{x}')dy \tag{20c}$$

$$= \delta_{(t,t)}\lambda_l + \epsilon_{\mathcal{S}_t}(\mathbf{x}'), \tag{20d}$$

where the inequality in (20c) is because:

$$\int_{\mathcal{Y}} (l_y^t - l_{y'}^t)p^t(y|\mathbf{x}')dy \leq |\mathbf{x} - \mathbf{x}'| \int_{\mathcal{Y}} \left|\frac{l_y^t - l_{y'}^t}{\mathbf{x} - \mathbf{x}'}\right| p^t(y|\mathbf{x}')dy \tag{21a}$$

$$\leq |\mathbf{x} - \mathbf{x}'| \int_{\mathcal{Y}} \lambda_l p^t(y|\mathbf{x}')dy \tag{21b}$$

$$\leq \delta_{(t,t)} \int_{\mathcal{Y}} \lambda_l p^t(y|\mathbf{x}')dy \tag{21c}$$

$$= \delta_{(t,t)}\lambda_l \int_{\mathcal{Y}} p^t(y|\mathbf{x}')dy \tag{21d}$$

$$= \delta_{(t,t)}\lambda_l, \tag{21e}$$

for which the equality in (21e) is because the integral of the density across the domain is 1:

$$\int_{\mathcal{Y}} p^t(y|\mathbf{x}')dy = 1. \tag{22}$$

The second term within the expectation in (19e) is bounded by:

$$\int_{\mathcal{Y}} l_y^t(p^t(y|\mathbf{x}) - p^t(y|\mathbf{x}'))dy \leq |\mathbf{x} - \mathbf{x}'| \int_{\mathcal{Y}} l_y^t \left|\frac{p^t(y|\mathbf{x}) - p^t(y|\mathbf{x}')}{\mathbf{x} - \mathbf{x}'}\right|)dy \tag{23a}$$

$$\leq \delta_{(t,t)}\lambda_t \int_{\mathcal{Y}} l_y^t dy \tag{23b}$$

$$= \frac{1}{3}\delta_{(t,t)}\lambda_t L_l^{\frac{3}{2}}. \tag{23c}$$

Combining all the inequalities together, we have:

$$\epsilon_F^t = \mathbb{E}_{\mathcal{X}^t} \left[ \int_{\mathcal{Y}} l_y^t p^t(y|\mathbf{x}')dy + \int_{\mathcal{Y}} l_y^t (p^t(y|\mathbf{x}) - p^t(y|\mathbf{x}'))dy \right] \tag{24a}$$

$$\leq \mathbb{E}_{\mathcal{X}^t} \left[ \epsilon_{\mathcal{S}_t}(\mathbf{x}') + \delta_{(t,t)}(\lambda_l + \frac{1}{3}\lambda_t L_l^{\frac{3}{2}}) \right] \tag{24b}$$

$$= \epsilon_{\mathcal{S}_t} + \delta_{(t,t)}(\lambda_l + \frac{1}{3}\lambda_t L_l^{\frac{3}{2}}), \tag{24c}$$

where the last equality holds due to $\mathbb{E}[\epsilon_{\mathcal{S}_t}(\mathbf{x}')] = \epsilon_{\mathcal{S}_t}$ and the invariance of constants under expectation, i,e., $\mathbb{E}(c) = c$ for any constant $c$.

Given that $\mu_{t=1} = p(t = 1)$ and $\mu_{t=0} = p(t = 0)$, where $\mu_{t=1} + \mu_{t=0} = 1$, we conclude the proof by expanding the expected factual loss $\epsilon_F$ by definition:

$$\epsilon_F = \int_{\mathcal{X} \times \mathcal{T}} l(\mathbf{x}, t) p(\mathbf{x}, t) d\mathbf{x}dt \tag{25a}$$

$$= \int_{\mathcal{X}} l^{t=1}(\mathbf{x}) p(\mathbf{x}, t = 1)d\mathbf{x} + \int_{\mathcal{X}} l^{t=0}(\mathbf{x}) p(\mathbf{x}, t = 0)d\mathbf{x} \tag{25b}$$

$$= \int_{\mathcal{X}} l^{t=1}(\mathbf{x}) p^{t=1}(\mathbf{x}) p(t = 1)d\mathbf{x} + \int_{\mathcal{X}} l^{t=0}(\mathbf{x}) p^{t=0}(\mathbf{x}) p(t = 0)d\mathbf{x} \tag{25c}$$

$$= p(t = 1) \cdot \mathbb{E}_{\mathcal{X}^{t=1}} \left[ (y - \hat{f}^{t=1}(\mathbf{x}))^2 \right] + p(t = 0) \cdot \mathbb{E}_{\mathcal{X}^{t=0}} \left[ (y - \hat{f}^{t=0}(\mathbf{x}))^2 \right] \tag{25d}$$

$$= \mu_{t=1} \cdot \mathbb{E}_{\mathcal{X}^{t=1}} \left[ \mathbb{E}_{\mathcal{Y}^{t=1}} \left[ (y - \hat{f}^{t=1}(\mathbf{x}))^2 \mid \mathbf{x} \right] \right] + \mu_{t=0} \cdot \mathbb{E}_{\mathcal{X}^{t=0}} \left[ \mathbb{E}_{\mathcal{Y}^{t=0}} \left[ (y - \hat{f}^{t=0}(\mathbf{x}))^2 \mid \mathbf{x} \right] \right] \tag{25e}$$

$$\leq \sum_{t \in \{0,1\}} \mu_t \left( \epsilon_{\mathcal{S}_t} + \delta_{(t,t)}(\lambda_l + \frac{1}{3}\lambda_t L_l^{\frac{3}{2}}) \right). \tag{25f}$$

**Discussion**: Lemma A.3 establishes a general upper bound on the factual loss under the core-set paradigm, which has been explored in prior studies (Sener & Savarese, 2018; Qin et al., 2021). Building on this foundation, we provide a more rigorous and comprehensive proof to strengthen the theoretical underpinnings. Also, we visualize the factual covering radius in Figure 1(a) and 1(c) to enhance interpretation, where the full coverage on the same class is required. □

### A.4. Proof of Lemma A.5

**Definition A.4.** *Let $\mathcal{H} = \{h|h : \mathcal{X} \to \mathbb{R}\}$ be a family of functions. The distribution distance measure – integral probability metric (IPM) between two data distributions $p^{t=1}(\mathbf{x})$ and $p^{t=0}(\mathbf{x})$ over the domain $\mathcal{X}$ is defined as:*

$$\text{IPM}_{\mathcal{H}}(p^{t=1}(\mathbf{x}), p^{t=0}(\mathbf{x})) = \sup_{h \in \mathcal{H}} \left| \int_{\mathcal{X}} h(\mathbf{x})(p^{t=1}(\mathbf{x}) - p^{t=0}(\mathbf{x}))d\mathbf{x} \right|. \tag{26}$$

**Lemma A.5.** *Denote the expected loss on subset $S_t$ as $\epsilon_{\mathcal{S}_t}$, the constant $\mu_t = p(t)$, counterfactual covering radius as $\delta_{(t,1-t)}$. Assume that the conditional probability density function $p^t(y|\mathbf{x})$ is $\lambda_t$-Lipschitz, the squared loss function $l$ is $\lambda_l$-Lipschitz and $l$ is further bounded by $L_l$. Also, let $\mathcal{H} = \{h|h : \mathcal{X} \to \mathbb{R}\}$ be a family of functions and $f : \mathcal{X} \times \mathcal{T} \to \mathcal{Y}$ be the hypothesis. Assume that there exists a constant $\kappa > 0$, such that $h_f(\mathbf{x}, t) := \frac{1}{\kappa} l_f(\mathbf{x}, t) \in \mathcal{H}$. The counterfactual expected loss is bounded as follows:*

$$\epsilon_{CF} \leq \sum_{t \in \{0,1\}} \mu_{1-t} \left( \epsilon_{\mathcal{S}_t} + \delta_{(t,1-t)}(\lambda_l + \frac{1}{3}\lambda_t L_l^{\frac{3}{2}}) \right) + \kappa_{\mathcal{H}}, \tag{27}$$

*where constant $\kappa_{\mathcal{H}} = \kappa \cdot IPM_{\mathcal{H}}(p^{t=1}(\mathbf{x}), p^{t=0}(\mathbf{x}))$ describes the distributional discrepancy between the treatment groups' distritbuions $(p^{t=1}(\mathbf{x})$ and $p^{t=0}(\mathbf{x}))$ over the domain $\mathcal{X}$, e.g., its i.i.d realization set $\mathcal{S}$ induced by AL.*

*Proof of Lemma A.5.* For notation simplicity, we denote $l_y^t = (y - \hat{f}^t(\mathbf{x}))^2$, and denote $p^{1-t}(\mathbf{x}) = p(\mathbf{x} \mid 1 - t)$, for

$\mathbf{x} \sim p^{1-t}(\mathbf{x})$, we have the counterfactual loss on the group $1 - t$ as:

$$\epsilon_{CF}^t = \mathbb{E}_{\mathcal{X}^{1-t}} \left[ (y - \hat{f}^t(\mathbf{x}))^2 \right] \tag{28a}$$

$$= \mathbb{E}_{\mathcal{X}^{1-t}} \left[ \mathbb{E}_{\mathcal{Y}^t} \left[ (y - \hat{f}^t(\mathbf{x}))^2 \mid \mathbf{x} \right] \right] \tag{28b}$$

$$= \mathbb{E}_{\mathcal{X}^{1-t}} \left[ \int_{\mathcal{Y}} l_y^t p^t(y|\mathbf{x}) dy \right] \tag{28c}$$

$$= \mathbb{E}_{\mathcal{X}^{1-t}} \left[ \int_{\mathcal{Y}} l_y^t \left( p^t(y|\mathbf{x}) - p^t(y|\mathbf{x}') + p^t(y|\mathbf{x}') \right) dy \right] \tag{28d}$$

$$= \mathbb{E}_{\mathcal{X}^{1-t}} \left[ \int_{\mathcal{Y}} l_y^t p^t(y|\mathbf{x}') dy + \int_{\mathcal{Y}} l_y^t \left( p^t(y|\mathbf{x}) - p^t(y|\mathbf{x}') \right) dy \right]. \tag{28e}$$

Note, that in (28d) we introduce the selected $\mathbf{x}' \in \mathcal{S}_t$ from the distribution $p^t(\mathbf{x})$ ($\mathbf{x}'$ is not directly sampled from $p^t(\mathbf{x})$ but from $\mathcal{S}_t$), which can be consider counterfactual sample w.r.t. $\mathbf{x} \sim p^{1-t}(\mathbf{x})$. That is, the sample $\mathbf{x}$ from the group $1 - t$ is covered by the counterfactual sample $\mathbf{x}' \in \mathcal{S}_t$ from group $t$ within the counterfactual radius $\delta_{(t,1-t)}$.

Similar to the deduction in Appendix A.3, the first term within the expectation in (28e) is bounded by:

$$\int_{\mathcal{Y}} l_y^t p^t(y|\mathbf{x}') dy = \int_{\mathcal{Y}} (l_y^t - l_{y'}^t + l_{y'}^t) p^t(y|\mathbf{x}') dy \tag{29a}$$

$$= \int_{\mathcal{Y}} (l_y^t - l_{y'}^t) p^t(y|\mathbf{x}') dy + \int_{\mathcal{Y}} l_{y'}^t p^t(y|\mathbf{x}') dy \tag{29b}$$

$$\leq \delta_{(t,1-t)} \lambda_l + \int_{\mathcal{Y}} l_{y'}^t p^t(y|\mathbf{x}') dy \tag{29c}$$

$$= \delta_{(t,1-t)} \lambda_l + \epsilon_{\mathcal{S}_t}(\mathbf{x}'). \tag{29d}$$

The inequality in (29c) is because:

$$\int_{\mathcal{Y}} (l_y^t - l_{y'}^t) p^t(y|\mathbf{x}') dy \leq |\mathbf{x} - \mathbf{x}'| \int_{\mathcal{Y}} \left| \frac{l_y^t - l_{y'}^t}{\mathbf{x} - \mathbf{x}'} \right| p^t(y|\mathbf{x}') dy \tag{29e}$$

$$\leq |\mathbf{x} - \mathbf{x}'| \int_{\mathcal{Y}} \lambda_l p^t(y|\mathbf{x}') dy \tag{29f}$$

$$\leq \delta_{(t,1-t)} \int_{\mathcal{Y}} \lambda_l p^t(y|\mathbf{x}') dy \tag{29g}$$

$$= \delta_{(t,1-t)} \lambda_l \int_{\mathcal{Y}} p^t(y|\mathbf{x}') dy \tag{29h}$$

$$= \delta_{(t,1-t)} \lambda_l. \tag{29i}$$

The equality in (29i) is because the integral of the density across the domain is 1:

$$\int_{\mathcal{Y}} p^t(y|\mathbf{x}') dy = 1. \tag{29j}$$

The second term within the expectation in (28e) is bounded by:

$$\int_{\mathcal{Y}} l_y^t \left( p^t(y|\mathbf{x}) - p^t(y|\mathbf{x}') \right) dy = |\mathbf{x} - \mathbf{x}'| \int_{\mathcal{Y}} l_y^t \frac{|p^t(y|\mathbf{x}) - p^t(y|\mathbf{x}')|}{|\mathbf{x} - \mathbf{x}'|} dy \tag{30a}$$

$$\leq \delta_{(t,1-t)} \lambda_t \int_{\mathcal{Y}} l_y^t dy \tag{30b}$$

$$= \frac{1}{3} \delta_{(t,1-t)} \lambda_t L_l^{\frac{3}{2}}. \tag{30c}$$

Combining all the inequalities together, we have:

$$\epsilon_{CF}^t = \mathbb{E}_{\mathcal{X}^{1-t}} \left[ \int_{\mathcal{Y}} l_y^t p^t(y|\mathbf{x}') dy + \int_{\mathcal{Y}} l_y^t \left( p^t(y|\mathbf{x}) - p^t(y|\mathbf{x}') \right) dy \right] \tag{31a}$$

$$\leq \mathbb{E}_{\mathcal{X}^{1-t}} [\epsilon_{\mathcal{S}_t}(\mathbf{x}')] + \delta_{(t,1-t)} \lambda_l + \frac{1}{3} \delta_{(t,1-t)} \lambda_t L_l^{\frac{3}{2}} \tag{31b}$$

$$= \mathbb{E}_{\mathcal{X}^{1-t}} [\epsilon_{\mathcal{S}_t}(\mathbf{x}')] + \delta_{(t,1-t)} \left( \lambda_l + \frac{1}{3} \lambda_t L_l^{\frac{3}{2}} \right), \tag{31c}$$

where the first term is the counterfactual loss on the subset $S_t$ induced by AL.

Extending the the expected counterfactual error $\epsilon_{CF}$ by definition:

$$\epsilon_{CF} = \int_{\mathcal{X} \times \mathcal{T}} l(\mathbf{x}, t) p(\mathbf{x}, 1 - t) d\mathbf{x} dt \tag{32a}$$

$$= \int_{\mathcal{X}} (y^{t=1} - \hat{f}^{t=1}(\mathbf{x}))^2 p(\mathbf{x}, t = 0) d\mathbf{x} + \int_{\mathcal{X}} (y^{t=0} - \hat{f}^{t=0}(\mathbf{x}))^2 p(\mathbf{x}, t = 1) d\mathbf{x} \tag{32b}$$

$$= \int_{\mathcal{X}} (y^{t=1} - \hat{f}^{t=1}(\mathbf{x}))^2 p^{t=0}(\mathbf{x}) p(t = 0) d\mathbf{x} + \int_{\mathcal{X}} (y^{t=0} - \hat{f}^{t=0}(\mathbf{x}))^2 p^{t=1}(\mathbf{x}) p(t = 1) d\mathbf{x} \tag{32c}$$

$$= p(t = 0) \cdot \mathbb{E}_{\mathcal{X}^{t=0}} \left[ (y^{t=1} - \hat{f}^{t=1}(\mathbf{x}))^2 \right] + p(t = 1) \cdot \mathbb{E}_{\mathcal{X}^{t=1}} \left[ (y^{t=0} - \hat{f}^{t=0}(\mathbf{x}))^2 \right] \tag{32d}$$

$$\leq \mu_{t=1} \left[ \mathbb{E}_{\mathcal{X}^{t=1}} [\epsilon_{\mathcal{S}_{t=0}}(\mathbf{x}')] + \delta_{(0,1)} \left( \lambda_l + \frac{1}{3} \lambda_{t=0} L_l^{\frac{3}{2}} \right) \right] + \mu_{t=0} \left[ \mathbb{E}_{\mathcal{X}^{t=0}} [\epsilon_{\mathcal{S}_{t=1}}(\mathbf{x}')] + \delta_{(1,0)} \left( \lambda_l + \frac{1}{3} \lambda_{t=1} L_l^{\frac{3}{2}} \right) \right] \tag{32e}$$

$$= (\epsilon_{CF})_{\mathcal{S}} + \mu_{t=1} \cdot \delta_{(0,1)} \left( \lambda_l + \frac{1}{3} \lambda_{t=0} L_l^{\frac{3}{2}} \right) + \mu_{t=0} \cdot \delta_{(1,0)} \left( \lambda_l + \frac{1}{3} \lambda_{t=1} L_l^{\frac{3}{2}} \right). \tag{32f}$$

Noticing that the counterfactual loss over the selected set $\mathcal{S}$ induced by AL: $(\epsilon_{CF})_{\mathcal{S}}$ in (32f) can be further bounded by the IPM, e.g., Wasserstein distance, by adapting the proof of "**Lemma 1**" of the main text in (Shalit et al., 2017) to the subset $\mathcal{S}$ induced by AL. Let $p(t = 1) = \mu$ be the marginal probability of treatment in $\mathcal{S}$, thus $p(t = 0) = 1 - \mu$ by noting that $p(t = 1) + p(t = 0) = 1$, we bound the first term in (32f) as follows:

$$(\epsilon_{CF})_{\mathcal{S}} - [(1 - \mu) \cdot (\epsilon_F)_{\mathcal{S}_{t=1}} + \mu \cdot (\epsilon_F)_{\mathcal{S}_{t=0}}] \tag{33a}$$

$$= [(1 - \mu) \cdot (\epsilon_{CF})_{\mathcal{S}_{t=1}} + \mu \cdot (\epsilon_{CF})_{\mathcal{S}_{t=0}}] - [(1 - \mu) \cdot (\epsilon_F)_{\mathcal{S}_{t=1}} + \mu \cdot (\epsilon_F)_{\mathcal{S}_{t=0}}] \tag{33b}$$

$$= (1 - \mu) \cdot [(\epsilon_{CF})_{\mathcal{S}_{t=1}} - (\epsilon_F)_{\mathcal{S}_{t=1}}] + \mu \cdot [(\epsilon_{CF})_{\mathcal{S}_{t=0}} - (\epsilon_F)_{\mathcal{S}_{t=0}}] \tag{33c}$$

$$= (1 - \mu) \int_{\mathcal{X}} l_x^{t=1} \left( p^{t=0}(\mathbf{x}) - p^{t=1}(\mathbf{x}) \right) d\mathbf{x} + \mu \int_{\mathcal{X}} l_x^{t=0} \left( p^{t=1}(\mathbf{x}) - p^{t=0}(\mathbf{x}) \right) d\mathbf{x} \tag{33d}$$

$$= (1 - \mu) \kappa \cdot \int_{\mathcal{X}} \frac{1}{\kappa} l_x^{t=1} \left( p^{t=0}(\mathbf{x}) - p^{t=1}(\mathbf{x}) \right) d\mathbf{x} + \mu \kappa \cdot \int_{\mathcal{X}} \frac{1}{\kappa} l_x^{t=0} \left( p^{t=1}(\mathbf{x}) - p^{t=0}(\mathbf{x}) \right) d\mathbf{x} \tag{33e}$$

$$\leq (1 - \mu) \kappa \cdot \sup_{h \in \mathcal{H}} \left| \int_{\mathcal{X}} h(x) \left( p^{t=0}(\mathbf{x}) - p^{t=1}(\mathbf{x}) \right) d\mathbf{x} \right| + \mu \kappa \cdot \sup_{h \in \mathcal{H}} \left| \int_{\mathcal{X}} h(x) \left( p^{t=1}(\mathbf{x}) - p^{t=0}(\mathbf{x}) \right) d\mathbf{x} \right| \tag{33f}$$

$$= \kappa \cdot \sup_{h \in \mathcal{H}} \left| \int_{\mathcal{X}} h(x) \left( p^{t=1}(\mathbf{x}) - p^{t=0}(\mathbf{x}) \right) d\mathbf{x} \right| = \kappa \cdot \text{IPM}_{\mathcal{H}}(p^{t=1}(\mathbf{x}), p^{t=0}(\mathbf{x})). \tag{33g}$$

Thus, noting that $(\epsilon_F)_{\mathcal{S}_{t=1}}$ is indeed the expected factual loss over the subset $\mathcal{S}_{t=1}$ induced by AL, thus simplying the notation to $\epsilon_{\mathcal{S}_t}$, we conclude the proof by plugging (33g) back to (32f):

$$\epsilon_{CF} \leq \sum_{t \in \{0,1\}} \mu_{1-t} \left( \epsilon_{\mathcal{S}_t} + \delta_{(t,1-t)} (\lambda_l + \frac{1}{3} \lambda_t L_l^{\frac{3}{2}}) \right) + \kappa_{\mathcal{H}}, \tag{34a}$$

where the constant $\kappa_{\mathcal{H}} = \kappa \cdot \text{IPM}_{\mathcal{H}}(p^{t=1}(\mathbf{x}), p^{t=0}(\mathbf{x}))$ is derived in the similar fashion in (Shalit et al., 2017).

$\square$

**Discussion**: Thus, we provide an adjustable bound depending on the size of the covering radius, where the infimum of the upper bound value is by labeling all the pool set to make the covering radius the least, the gap originates from the labeling less data but with some covering radius. Note, that generalization bound by Shalit et al. (2017) quantifies well the risk upper bound given the fully labeled pool set. The work (Qin et al., 2021) builds upon (Shalit et al., 2017), but certain aspects remain underexplored, that is, it does not show the importance of the counterfactual covering radius. We fill this crucial theoretical gap by providing an informative and complete bound under the data-expanding context from scratch to derive a new bound tailored specifically to treatment effect estimation with AL. Furthermore, Lemma A.5 unveils the distinctive *counterfactual covering radius* which originates from the unique nature of the counterfactual prediction. We visualize the counterfactual covering radius $\delta_{(t,1-t)}$ in Figure 1(b) and 1(d) to facilitate the conceptualization of Lemma A.5 in the treatment effect estimation setting under AL paradigm, where the full coverage on the counterfactual class is required.

### A.5. Proof of Theorem 4.1

**Definition A.6.** Let $\mathcal{S}_{(t,t)}^*$ of size $B_{(t,t)}$ denote the optimal (OPT) subset for treatment group $t$. For each point $v^t \in \mathcal{S}_{(t,t)}^*$, let the cluster of $v^t$ be $\mathcal{C}_{v^t}^* = \{u^t \in \mathcal{D}_t : d(u^t, v^t) = \min_{v' \in \mathcal{S}_{(t,t)}^*} d(u^t, v')\}$. As such, we have partitions $\mathcal{C}_{v_1^t}^*, \mathcal{C}_{v_2^t}^*, \ldots, \mathcal{C}_{v_{B_{(t,t)}}^t}^*$, where each point $u^t \in \mathcal{D}_t$ is placed in the closest $\mathcal{C}_{v_i^t}^*$ w.r.t. $v_i^t \in \mathcal{S}_{(t,t)}^*$.

**Theorem 4.1.** *Under Assumption 2.3, the sum of the covering radii returned by Algorithm 1 is upper-bounded by* $2 \times \sum_{t\{0,1\}}(OPT_{\delta_{(t,t)}} + OPT_{\delta_{(t,1-t)}})$

*Proof of Theorem 4.1.* Let the output of the Algorithm 1 be $\mathcal{S}$, specifically, the total budget $B$, additive arithmetically, is split into four parts with $B = B_{(1,1)} + B_{(1,0)} + B_{(0,0)} + B_{(0,1)}$, i.e., with each part to acquire designated point to reduce one of the four radius $\delta_{(1,1)}$, $\delta_{(1,0)}$, $\delta_{(0,0)}$, and $\delta_{(0,1)}$ at a time. Thus, we have $\mathcal{S} = \mathcal{S}_{(1,1)} \cup \mathcal{S}_{(1,0)} \cup \mathcal{S}_{(0,0)} \cup \mathcal{S}_{(0,1)}$, and noting that $\mathcal{S}_1 = \mathcal{S}_{(1,1)} \cup \mathcal{S}_{(1,0)}$, and $\mathcal{S}_0 = \mathcal{S}_{(0,0)} \cup \mathcal{S}_{(0,1)}$. To bound each of the radius:

- For $u^{t=1} \in \mathcal{D}_1$ to reduce $\delta_{(1,1)}$:

$$d(u^{t=1}, \mathcal{S}_1) \leq d(u^{t=1}, \mathcal{S}_{(1,1)}) \leq 2 \times OPT_{\delta_{(1,1)}}, \tag{35}$$

  where the first inequality is because $\mathcal{S}_{(1,1)} \subset \mathcal{S}_1$, and the second inequality is by Lemma A.8.

- For $u^{t=0} \in \mathcal{D}_0$ to reduce $\delta_{(1,0)}$ :

$$d(u^{t=0}, \mathcal{S}_1) \leq d(u^{t=0}, \mathcal{S}_{(1,0)}) \leq 2 \times OPT_{\delta_{(1,0)}}, \tag{36}$$

  where the first inequality is because $\mathcal{S}_{(1,0)} \subset \mathcal{S}_1$, and the second inequality is by Lemma A.9.

- For $u^{t=0} \in \mathcal{D}_0$ to reduce $\delta_{(0,0)}$ :

$$d(u^{t=0}, \mathcal{S}_0) \leq d(u^{t=0}, \mathcal{S}_{(0,0)}) \leq 2 \times OPT_{\delta_{(0,0)}}, \tag{37}$$

  where the first inequality is because $\mathcal{S}_{(0,0)} \subset \mathcal{S}_0$, and the second inequality is by Lemma A.8.

- For $u^{t=1} \in \mathcal{D}_1$ to reduce $\delta_{(0,1)}$ :

$$d(u^{t=1}, \mathcal{S}_0) \leq d(u^{t=0}, \mathcal{S}_{(0,1)}) \leq 2 \times OPT_{\delta_{(0,1)}}, \tag{38}$$

  where the first inequality is because $\mathcal{S}_{(0,1)} \subset \mathcal{S}_0$, and the second inequality is by Lemma A.9.

Since for all $u$ the above holds, thus it follows that

$$\max_{u \in \mathcal{D}_1 \backslash \mathcal{S}_1} d(u^{t=1}, \mathcal{S}_1) + \max_{u \in \mathcal{D}_0 \backslash \tilde{\mathcal{S}}_0} d(u^{t=0}, \mathcal{S}_1) + \max_{u \in \mathcal{D}_0 \backslash \mathcal{S}_0} d(u^{t=0}, \mathcal{S}_0) + \max_{u \in \mathcal{D}_1 \backslash \tilde{\mathcal{S}}_1} d(u^{t=1}, \mathcal{S}_0) \tag{39a}$$

$$\leq 2 \times OPT_{\delta_{(1,1)}} + 2 \times OPT_{\delta_{(1,0)}} + 2 \times OPT_{\delta_{(0,0)}} + 2 \times OPT_{\delta_{(0,1)}} \tag{39b}$$

$$\tag{39c}$$

$$\implies \delta_{(1,1)} + \delta_{(1,0)} + \delta_{(0,0)} + \delta_{(0,1)} \leq 2 \times \sum_{t\{0,1\}} \left( \text{OPT}_{\delta_{(t,t)}} + \text{OPT}_{\delta_{(t,1-t)}} \right) \tag{39d}$$

$\square$

In the following, we define $d(u, \mathcal{Q}) := \min_{v \in \mathcal{Q}} d(u, v)$ where $\mathcal{Q}$ can be any set, e.g., the optimal solution $\mathcal{S}^*_{(t,t)}$, also we denote the $OPT$ as the minimal covering radius returned by the optimal set $\mathcal{S}^*$, i.e., $OPT = \max_{u \in \mathcal{D}} \min_{v \in \mathcal{S}^*} d(u, v)$.

**Lemma A.7.** *Without loss of generality to Definition A.6, we have $\forall u, w \in \mathcal{C}_v$ for $v \in \mathcal{Q}$, then $d(u, w) \leq 2 \times OPT$*

*Proof of Lemma A.7.*

$$d(u, w) \leq d(u, v) + d(w, v) \tag{40a}$$
$$= d(u, \mathcal{Q}) + d(w, \mathcal{Q}) \tag{40b}$$
$$\leq OPT + OPT = 2 \times OPT, \tag{40c}$$

where the inequality (40a) is by the triangular inequality, the equality (40b) is by Definition A.6, and we complete the proof with the inequality (40c) due to $d(u, \mathcal{Q}) = \min_{v \in \mathcal{Q}} d(u, v) \leq \max_{u \in \mathcal{D}} \min_{v \in \mathcal{Q}} d(u, v) = OPT$. $\square$

Note, that the minimal covering radius $\delta_{(\cdot, \cdot)}$ under the optimal solution $\mathcal{S}^*_{(\cdot, \cdot)}$ is denoted as $\text{OPT}_{\delta_{(\cdot, \cdot)}}$.

**Lemma A.8.** *With budget $B_{(t,t)}$ Subset $S_{(t,t)}$ for the treatment group $t$ returned by Algorithm 1 is a $2 - OPT_{\delta_{(t,t)}}$ for covering the set $\mathcal{D}_t$ of the treatment group $t$.*

*Proof of Lemma A.8.* When covering the treatment group $t$, we dose not have any assumption for the data distribution, thus we split the proof into two scenarios in a similar fashion by Dinitz (2019), i.e., $\forall v \in \mathcal{S}^*_{(t,t)}, \mathcal{S}_{(t,t)} \cap \mathcal{C}^*_v \neq \varnothing$ and $\exists v \in \mathcal{S}^*_{(t,t)}, \mathcal{S}_{(t,t)} \cap \mathcal{C}^*_v = \varnothing$:

- $\forall v \in \mathcal{S}^*_{(t,t)}, \mathcal{S}_{(t,t)} \cap \mathcal{C}^*_v \neq \varnothing$: Let $w \in \mathcal{S}_{(t,t)} \cap \mathcal{C}^*_v$, for $u \in \mathcal{D}_t$, let $u \in \mathcal{C}^*_v$ with $v \in \mathcal{S}^*_{(t,t)}$ (noting that $\mathcal{C}^*_v \subset \mathcal{D}_t$), we have:

$$d(u, \mathcal{S}_{(t,t)}) \leq d(u, w) \leq 2 \times \text{OPT}_{\delta_{(t,t)}} \tag{41}$$

  where the first inequality is because $w \in \mathcal{S}_{(t,t)}$, and the second inequality is by Lemmma A.7 for $u, w \in \mathcal{C}^*_v$. Note, that Eq. (41) holds $\forall u \in \mathcal{D}_t$ as $\mathcal{C}^*_v \subset \mathcal{D}_t$, then $\delta_{(t,t)} = \max_{u \in \mathcal{D}} d(u, \mathcal{S}_{(t,t)}) \leq 2 \times \text{OPT}_{\delta_{(t,t)}}$.

- $\exists v \in \mathcal{S}^*_{(t,t)}, \mathcal{S}_{(t,t)} \cap \mathcal{C}^*_v = \varnothing$: Since $|\mathcal{S}_{(t,t)}| = |\mathcal{S}^*| = B_{(t,t)}$, by the pigeonhole principle, $\exists v' \in \mathcal{S}^*_{(t,t)}, s.t. |\mathcal{S}_{(t,t)} \cap \mathcal{C}^*_{v'}| \geq 2$. Thus, assume that for $z, m \in \mathcal{S}_{(t,t)} \cap \mathcal{C}^*_{v'}$ and $z$ is added to $\mathcal{S}_{(t,t)}$ before $m$. Let $\mathcal{S}'_{(t,t)} = \mathcal{S}_{(t,t)} \backslash m$, then for $u \in \mathcal{D}_t$, we have:

$$d(u, \mathcal{S}_{(t,t)}) \leq d(u, \mathcal{S}'_{(t,t)}) \tag{42a}$$
$$\leq d(m, \mathcal{S}'_{(t,t)}) \tag{42b}$$
$$\leq d(m, z) \tag{42c}$$
$$\leq 2 \times \text{OPT}_{\delta_{(t,t)}}, \tag{42d}$$

  where inequality in (42a) is because $\mathcal{S}'_{(t,t)} \subset \mathcal{S}_{(t,t)}$ (the radius decreases monotonically with larger set), the inequality in (42b) is because the greedy selection with larger distance with set $\mathcal{S}'_{(t,t)}$ ($m$ is selected before $u$), the inequality in (42c) is by definition that $d(m, \mathcal{S}'_{(t,t)}) = \min_{v \in \mathcal{S}'_{(t,t)}} d(m, v)$ and $z \in \mathcal{S}'_{(t,t)}$, the last inequality in (42d) is by Lemma A.7.

Since for all $u \in \mathcal{D}_t$, we have $d(u, \mathcal{S}_{(t,t)}) \leq 2 \times OPT_{\delta_{(t,t)}}$, then it follows that $\max_{u \in \mathcal{D}_t} d(u, \mathcal{S}_{(t,t)}) \leq 2 \times OPT_{\delta_{(t,t)}}$ to conclude the proof for A.8 by enumerating all the scenarios.

$\square$

**Lemma A.9.** *With budget $B_{(t,1-t)}$ Subset $\mathcal{S}_{(t,1-t)}$ for the treatment group $t$ returned by Algorithm 1 is a $2 - OPT_{\delta_{(t,1-t)}}$ for covering the set $\mathcal{D}_{1-t}$ of the treatment group $1 - t$.*

*Proof of Lemma A.9.* Note, that the reduction of the counterfactual covering radius $\delta_{(t,1-t)}$ requires the acquisition from group $t$, which is complex because the radius $\delta_{(t,1-t)}$ is calculated by $\delta_{(t,1-t)} = \max_{i\in\mathcal{D}_{1-t}\backslash\tilde{S}_{1-t}} \min_{j\in\mathcal{S}_{(t,1-t)}} d(\mathbf{x}_i^{1-t}, \mathbf{x}_j^t)$, noting that the proxy collection $\tilde{S}_{1-t}$ is from the treatment group $1-t$, however, the acquisition targeting querying the sample from group $t$ to reduce the counterfactual covering radius $\delta_{(t,1-t)}$.

Thus, by the greedy nature of the Algorithm 1, when $\delta_{(t,1-t)}$ is the one to be reduced, the mentality for the query step is to calculate the proxy point $a_{1-t} = \arg\max_{i\in\mathcal{D}_{1-t}\backslash\tilde{s}_{1-t}} \min_{j\in\mathcal{S}_{(t,1-t)}} d(\mathbf{x}_i^{1-t}, \mathbf{x}_j^t)$ from group $1-t$, then find the nearest point $a_t \in \mathcal{D}_t$ to $a_{1-t}$ as the factual query to expand $\mathcal{S}_{(t,1-t)}$. If there always exists $d(a_t, a_{1-t}) = 0$, the radius reduction can be real quick.

**Definition A.10.** Let cluster $\mathcal{F}_{v^t}^* = \{u^{1-t} \in \mathcal{D}_{1-t} : d(u^{1-t}, v^t) = \min_{v'\in\mathcal{S}_{(t,1-t)}^*} d(u^{1-t}, v')\}$ for $v^t \in \mathcal{S}_{(t,1-t)}^*$.

The proof adopts the similar mentality as shown in Proof of Lemma A.8, For now we show the proof for the first scenario and the second scenarios follows.

$\forall v \in \mathcal{S}_{(t,1-t)}^*, \tilde{S}_{1-t} \cap \mathcal{F}_v^* \neq \varnothing$: Let $w \in \tilde{S}_{1-t} \cap \mathcal{F}_v^*$, let $u \in \mathcal{F}_v^*$ with $v \in \mathcal{S}_{(t,1-t)}^*$ (noting that $\mathcal{F}_v^* \subset \mathcal{D}_{1-t}$), we have:

$$d(u^{1-t}, \mathcal{S}_{(t,1-t)}) = d(u^{1-t}, \tilde{S}_{1-t}) \tag{43a}$$
$$\leq d(u^{1-t}, w^{1-t}) \tag{43b}$$
$$\leq d(u^{1-t}, v^t) + d(u^{1-t}, v^t) \tag{43c}$$
$$\leq d(u^{1-t}, \mathcal{S}_{(t,1-t)}^*) + d(w^{1-t}, \mathcal{S}_{(t,1-t)}^*) \tag{43d}$$
$$\leq 2 \times \text{OPT}_{\delta_{(t,1-t)}}, \tag{43e}$$

where the equality (43a) is by the Assumption (Strong Ignorability) 2.3 (i.e., $0 < p(t|\mathbf{x}) < 1$), s.t., for $\tilde{S}_{1-t}$ returned by Algorithm 1, we can have identical set $\mathcal{S}_{(t,1-t)} \in \mathcal{D}_t$ to the proxy collection $\tilde{S}_{1-t}$, the inequality in (43b) is because $w^{1-t} \in \tilde{S}_{1-t}$, the inequality (43c) is by the triangular inequality, the inequality (43d) is by Definition A.10, and inequality (43e) is by $d(u^{1-t}, \mathcal{S}_{(t,1-t)}^*) = \min_{v\in\mathcal{S}_{(t,1-t)}^*} d(u^{1-t}, v) \leq \max_{u^{1-t}\in\mathcal{D}_{1-t}} \min_{v\in\mathcal{S}_{(t,1-t)}^*} d(u^{1-t}, v) = OPT_{\delta_{(t,1-t)}}$. $\square$

## A.6. Proof of Theorem 4.4

**Assumption 4.3** *Given the fixed covering radius $\delta_{(t,t)}$ and $\delta_{(t,1-t)}$, there exists the optimal solution $\mathcal{S}_t^*$, $\mathcal{S}_t^* \subset \mathcal{S}^*$ for treatment group $t$, such that $\mathcal{A}_F^{t=1} \bigcup \mathcal{A}_{CF}^{t=1} = \mathcal{D}$.*

**Definition A.11.** Let the pool set be $\mathcal{D}$ of size $n$, $\mathcal{S}_t$ of size $B_t$ be the solution for group $t$ returned by Algorithm 2. A family of sets $\mathcal{U} = \{\mathcal{U}_i\}_{i=v_1}^{v_m}$, where $\forall i, \mathcal{U}_i \subset \mathcal{D}$. Note, that the subscript $i$ of $\mathcal{U}_i$ denotes a single selected point $v_i$ in $\mathcal{S}_t \subseteq [v_m]$ due to the fact that each $v_i$ as the center with the fixed radius covers a set of point, i.e., by definition $\mathcal{U}_i = \mathcal{A}_{(t,t)}(v_i) \cup \mathcal{A}_{(t,1-t)}(v_i)$. Let the uncovered set be $\Omega_r = \mathcal{D}\backslash\bigcup_{i\in\mathcal{S}_t}\mathcal{U}_i$ up to iteration $r$, and assume that Algorithm 2 were to pick $\mathcal{U}_1', \mathcal{U}_2', ..., \mathcal{U}_k'$, for which $\mathcal{U}_i'$ is one of the set in $\mathcal{U}$. Denote the set covered by optimal solution $\mathcal{S}_t^*$ as $\Theta$, $\omega_r = |\mathcal{U}_r' \cap \Omega_{r-1}|$, and $\eta_i = |\Theta| - \sum_{j\leq i} \omega_j$.

**Theorem 4.4.** *Under Assumption 4.3, Algorithm 2 is a $(1 - \frac{1}{e})$ – approximation for the full coverage constraint on the equally weighted graph and unscaled out-degree.*

*Proof of Theorem 4.4.* We prove the $(1 - \frac{1}{e})$−approximation for the full coverage constraint by Algorithm 2 by extending the method by Dinitz (2019) for solving the conventional Max $k$−Cover Problem into our *Factual and Counterfactual Coverage Maximization* for the data-efficient treatment effect estimation problem. The objective is to query $\mathcal{S} = \mathcal{S}_1 \cup \mathcal{S}_0$ that maximizes the mean coverage $P(\mathcal{A})$.

As defined in Section 4.2, $\mathcal{A}_F^{t=1} = \bigcup_{\mathbf{x}\in\mathcal{S}_1} \mathcal{A}_{(1,1)}(\mathbf{x})$ and $\mathcal{A}_{CF}^{t=1} = \bigcup_{\mathbf{x}\in\mathcal{S}_1} \mathcal{A}_{(1,0)}(\mathbf{x})$, by further in Definition A.11, for the

factual ($\mathcal{A}_F^{t=1}$) and counterfactual ($\mathcal{A}_{CF}^{t=1}$) covered region by $S_1$, we have:

$$\mathcal{A}_F^{t=1} \cup \mathcal{A}_{CF}^{t=1} = |\bigcup_{j \leq B_t} \mathcal{U}_j'| \tag{44a}$$

$$= \sum_{j \leq B_t} \omega_j \tag{44b}$$

$$= |\Theta| - \eta_{B_t} \tag{44c}$$

$$\geq |\Theta| - |\Theta| e^{-1} = (1 - e^{-1}) |\Theta|, \tag{44d}$$

where the equality in (44a) is by definition of the covered region in Section 4.2 and the set $\mathcal{U}_i$ in Definition A.11, the equality in (44b) and (44c) is by Definition A.11, and the inequaltiy in (44d) is by Lemma A.14.

Under Assumption 4.3, there exists optimal solution s.t. $\Theta = \mathcal{D}$, which further implies:

$$\mathcal{A}_F^{t=1} \cup \mathcal{A}_{CF}^{t=1} \geq (1 - e^{-1})|\mathcal{D}| \implies P(\mathcal{A}_F^{t=1} \cup \mathcal{A}_{CF}^{t=1}) \geq \frac{(1 - e^{-1})|\mathcal{D}|}{|\mathcal{D}|} = 1 - e^{-1}. \tag{45}$$

Without loss of generality, the proof above applies for proving that

$$P(\mathcal{A}_F^{t=0} \cup \mathcal{A}_{CF}^{t=0}) \geq 1 - e^{-1}. \tag{46}$$

To conclude, as we have:

$$P(\mathcal{A}) = \frac{1}{4} P(\mathcal{A}_F^{t=1}) + \frac{1}{4} P(\mathcal{A}_{CF}^{t=1}) + \frac{1}{4} P(\mathcal{A}_F^{t=0}) + \frac{1}{4} P(\mathcal{A}_{CF}^{t=0}) \tag{47a}$$

$$= \frac{1}{2} P(\mathcal{A}_F^{t=1} \cup \mathcal{A}_{CF}^{t=1}) + \frac{1}{2} P(\mathcal{A}_F^{t=0} \cup \mathcal{A}_{CF}^{t=0}) \tag{47b}$$

$$\geq 1 - e^{-1}, \tag{47c}$$

where the equality in (47a) is by the definition for the mean coverage $P(\mathcal{A})$ and note that our maximization goal in Eq. (10) leave out the constant coefficient $1/4$, which does not affect the ultimate goal. The equality in (47b) is by the independence between the factual covering and counterfactual covering, the inequality in (47c) is by conclusion in Eq. (45) and (46). □

**Lemma A.12.** *For set $\mathcal{P}$, $\mathcal{Q}$ and let $|\mathcal{P}| \geq |\mathcal{Q}|$, we have $|\mathcal{P} \backslash \mathcal{Q}| \geq |\mathcal{P}| - |\mathcal{Q}|$.*

*Proof of Lemma A.12.* $|\mathcal{P} \backslash \mathcal{Q}| = |\mathcal{P}| - |\mathcal{P} \cap \mathcal{Q}| \geq |\mathcal{P}| - |\mathcal{Q}|$ due to the fact that $|\mathcal{P} \cap \mathcal{Q}| \leq |\mathcal{Q}|$ for $|\mathcal{P}| \geq |\mathcal{Q}|$. □

**Lemma A.13.** $\eta_i \leq \left| \Theta \backslash \bigcup_{j \leq i} \mathcal{U}_j' \right|.$

*Proof of Lemma A.13.*

$$\eta_i = |\Theta| - \sum_{j \leq i} \omega_j = |\Theta| - |\bigcup_{j \leq i} \mathcal{U}_j'| \leq |\Theta \backslash \bigcup_{j \leq i} \mathcal{U}_j'|. \tag{48}$$

The first and second equality is straightforward by definition and observation, and the inequality is by Lemma A.12 due to the fact that the optimal cover $\Theta$ is the larger. □

**Lemma A.14.** $\eta_{B_t} \leq |\Theta| e^{-1}.$

*Proof of Lemma A.14.* Construct the subtraction:

$$\eta_i - \eta_{i-1} = |\Theta| - \sum_{j \leq i} \omega_j - \left( |\Theta| - \sum_{j \leq i-1} \omega_j \right) = -\omega_i \tag{49a}$$

$$\implies \eta_i = \eta_{i-1} - \omega_i \tag{49b}$$

Lemma A.13 implies that optimal solution $\Theta$ covers at least $\eta_i$ uncovered samples (the uncovered samples are w.r.t. to the covered set by $\mathcal{S}_t$) with $B_t$ sets, thus by the pigeonhole principle, when $\eta_i$ filled into $B_t$ sets, each set assigned averagely $\eta_i/B_t$, however, there are less than $B_t$ sets availables for the Algorithm 2 to query, thus there exists the set covers at least $\eta_i/B_t$ uncovered samples. By the greedy nature of the Algorithm 2, we have $\omega_{i+1} \geq \frac{\eta_i}{B_t}$, then it further implies that:

$$\eta_i = \eta_{i-1} - \omega_i \leq \eta_{i-1} - \frac{\eta_{i-1}}{B_t} = \eta_{i-1}(1 - \frac{1}{B_t}) \tag{50a}$$

$$\implies \frac{\eta_i}{\eta_{i-1}} = \frac{1}{B_t} \tag{50b}$$

For $\omega_0 = 0$ implies that $\eta_0 = |\Theta|$, and maximally with budget $B_t$, we have $\eta_{B_t}$, and then construct the following:

$$\frac{\eta_{B_t}}{\eta_0} = \underbrace{\frac{\eta_{B_t}}{\eta_{B_t-1}} \times \frac{\eta_{B_t-1}}{\eta_{B_t-2}} \times \cdots \times \frac{\eta_2}{\eta_1} \times \frac{\eta_1}{\eta_0}}_{B_t \text{ quantities above}} = (1 - \frac{1}{B_t})^{B_t} \tag{51}$$

Noting that:

$$\lim_{B_t \to \infty} (1 - \frac{1}{B_t})^{B_t} = \frac{1}{e} \implies \frac{\eta_{B_t}}{\eta_0} \leq \frac{1}{e}. \tag{52}$$

Thus, it can be concluded that $\eta_{B_t} \leq |\Theta| e^{-1}$. $\square$

# B. Related Work

**Treatment effect estimation.** Many early works in causal effect estimation (a.k.a., the treatment effect estimation) focus on group-level estimation, e.g., conditional average treatment effect (CATE). The widely used inverse probability weighting method (Rosenbaum & Rubin, 1983; Imbens & Rubin, 2015) and the doubly robust model (Robins et al., 1994) are designed to mitigate the selection bias in CATE estimation, but are not generalizable to the unseen individuals or groups without labels. So far, various methods (Shalit et al., 2017; Louizos et al., 2017; Alaa & Van Der Schaar, 2017; Yao et al., 2018; Yoon et al., 2018; Shi et al., 2019; Zhang et al., 2020; Kallus, 2020; Jesson et al., 2020; Wang et al., 2024) have been proposed due to the proliferation of deep learning (DL). These parametric models are good at modeling the individual-level causal effect and are generalizable to unseen instances. Furthermore, the strong expressive power of such deep models can handle the high-dimensional data and relax the pivot assumptions in causal effect estimation, e.g., unconfoundedness assumption, by learning the deconfounded latent representations via neural mapping for treated and control groups. Additionally, there is another branch of work that investigate the treatment effect estimation under interference (e.g., the violation of the SUTVA assumptions) (Rakesh et al., 2018; Ma & Tresp, 2021; Ma et al., 2022; Lin et al., 2023; 2024; Chen et al., 2024; Lin et al., 2025), which is out-of-scope to the foucs of this paper.

**Active learning.** The concept of active learning (AL) dates back over a century (Smith, 1918). Over time, it has evolved into a prominent branch of machine learning research (Settles, 2009; Ren et al., 2021; Zhan et al., 2022). The primary goal of AL is to optimize model performance in a cost-efficient manner, achieving low model risk while minimizing the number of labeled samples required. AL methods are typically categorized into three main scenarios: query synthesis (Wang et al., 2015), stream-based (Fujii & Kashima, 2016), and pool-based approaches (Wu, 2018). This paper focuses on pool-based AL, particularly in regression problems, where key acquisition strategies include uncertainty-based sampling (Gal et al., 2017), density-based querying (Sener & Savarese, 2018), and hybrid methods (Ash et al., 2019). For example, Bayesian Active Learning by Disagreement (BALD) (Gal et al., 2017) uses epistemic uncertainty to select unlabeled samples, while core-set (Sener & Savarese, 2018) prioritizes samples based on their maximum distance to the nearest neighbor in the hidden space. ACS-FW (Pinsler et al., 2019) combines core-set and Bayesian approaches, balancing sample diversity and uncertainty in batch-mode acquisition. Although general AL methods are not specifically designed for CEE, benchmarking these methods can yield valuable insights.

**Treatment effect estimation with active learning.** Thus far, some progress has been made in this area of research. For instance, (Sundin et al., 2019) proposes a querying criterion based on the estimated S-type error rate—the probability that the model incorrectly infers the sign of the treatment effect. However, this work focuses on estimating the correct sign of the treatment effect, which differs from the risk metric used in our study. For research aligned with the same risk metric,

Qin et al. (2021) introduce a theoretical framework that extends the upper-bound formulation from (Shalit et al., 2017) mainly by a core-set approach (Tsang et al., 2005). Despite this, their proposed algorithm QHTE does not adequately address distribution alignment during data acquisition. To mitigate acquisition imbalance, Causal-BALD (Jesson et al., 2021b) adopts an information-theoretic perspective, introducing the $\mu\rho$BALD criterion. This criterion scales the acquisition metric inversely with counterfactual variance, encouraging the selection of samples that align with similar counterfactuals when certain treatments are underrepresented. This represents a notable improvement over its predecessor, $\mu$BALD, an uncertainty-based softmax-BALD method (Kirsch et al., 2023). However, Causal-BALD depends heavily on accurate uncertainty quantification and computationally intensive training using complex estimators, such as deep kernel learning (Wilson et al., 2016). Recently, Wen et al. (2025) introduced a straightforward yet effective algorithm, MACAL, which reduces distributional discrepancies while remaining model-independent. Despite its advantages, MACAL requires querying data in pairs (one from the treated group and one from the control group), limiting its generalizability in scenarios where optimality can be achieved by querying from only one treatment group. Furthermore, while MACAL includes convergence analysis for sub-objectives, it lacks guarantees on overall risk upper-bound convergence. Additionally, some studies (Deng et al., 2011; Addanki et al., 2022; Connolly et al., 2023; Ghadiri et al., 2024) leverage AL for efficient experimental trial design, where treatment information is applied only after sample acquisition, rather than being included in the initial pool. This setup differs fundamentally from our focus, where treatment information is available from the start. Besides, the recent exploration of AL for treatment effect estimation on the feature level (Piskorz et al., 2025) differs from our individual-level data acquisition task.

---

**Algorithm 1** Greedy Radius Reduction

---

1: **Input:** Pool set $D = \mathcal{D}_1 \cup \mathcal{D}_0$; random initialization $\mathcal{S}^{\text{init}} = \mathcal{S}_1 \cup \mathcal{S}_0$, where $\mathcal{S}_1$ and $\mathcal{S}_0$ are the random initialization set for the treated and control group respectively; budget $B$; distance metric $d(\cdot, \cdot)$

2: $\quad \mathcal{S} \leftarrow \mathcal{S}^{\text{init}}, \tilde{S}_1 \leftarrow \varnothing, \tilde{S}_0 \leftarrow \varnothing$

3: **while** $|\mathcal{S}| < |\mathcal{S}^{\text{init}}| + B$ **do**

4: $\quad\quad \delta \leftarrow \max\{\delta_{(1,1)}, \delta_{(1,0)}, \delta_{(0,0)}, \delta_{(0,1)}\}$

5: $\quad\quad$ **if** $\delta == \delta_{(1,1)}$ **then**

6: $\quad\quad\quad a \leftarrow \arg\max_{i \in \mathcal{D}_1 \backslash \mathcal{S}_1} \min_{j \in \mathcal{S}_1} d(\mathbf{x}_i^{t=1}, \mathbf{x}_j^{t=1})$ $\quad$ {To reduce $\delta_{(1,1)}$ by querying data point from group $t = 1$}

7: $\quad\quad$ **else if** $\delta == \delta_{(0,0)}$ **then**

8: $\quad\quad\quad a \leftarrow \arg\max_{i \in \mathcal{D}_0 \backslash \mathcal{S}_0} \min_{j \in \mathcal{S}_0} d(\mathbf{x}_i^{t=0}, \mathbf{x}_j^{t=0})$ $\quad$ {To reduce $\delta_{(0,0)}$ by querying data point from group $t = 0$}

9: $\quad\quad$ **else if** $\delta == \delta_{(1,0)}$ **then**

10: $\quad\quad\quad a' \leftarrow \arg\max_{i \in \mathcal{D}_0 \backslash \tilde{S}_0} \min_{j \in \mathcal{S}_1} d(\mathbf{x}_i^{t=0}, \mathbf{x}_j^{t=1})$ $\quad$ {To reduce $\delta_{(1,0)}$ by querying data point from group $t = 0$}

11: $\quad\quad\quad b \leftarrow \arg\min_{i \in \mathcal{D}_1} d(\mathbf{x}_i^{t=1}, a')$ $\quad$ {Find the nearest point from $t = 1$ to $a'$ to reduce counterfactual radius $\delta_{(1,0)}$}

12: $\quad\quad\quad$ **if** $b \notin \mathcal{S}_1$ **then**

13: $\quad\quad\quad\quad a \leftarrow b, \tilde{S}_0 \leftarrow \tilde{S}_0 \cup \{a'\}$ $\quad$ {Counterfactual radius $\delta_{(1,0)}$ can be reduced, add $b$ into the training set}

14: $\quad\quad\quad$ **else**

15: $\quad\quad\quad\quad$ Repeat Line 4-25 by excluding $\delta_{(1,0)}$ from the $\max$ function $\quad$ {Counterfactual radius $\delta_{(1,0)}$ cannot be reduced, go back to other reducible covering radii from the largest one}

16: $\quad\quad\quad$ **end if**

17: $\quad\quad$ **else if** $\delta == \delta_{(0,1)}$ **then**

18: $\quad\quad\quad a' \leftarrow \arg\max_{i \in \mathcal{D}_1 \backslash \tilde{S}_1} \min_{j \in \mathcal{S}_0} d(\mathbf{x}_i^{t=1}, \mathbf{x}_j^{t=0})$ $\quad$ {To reduce $\delta_{(0,1)}$ by querying data point from group $t = 1$}

19: $\quad\quad\quad b \leftarrow \arg\min_{i \in \mathcal{D}_0} d(\mathbf{x}_i^{t=0}, a')$ $\quad$ {Find the nearest point from $t = 0$ to $a'$ to reduce counterfactual radius $\delta_{(0,1)}$}

20: $\quad\quad\quad$ **if** $b \notin \mathcal{S}_0$ **then**

21: $\quad\quad\quad\quad a \leftarrow b, \tilde{S}_1 \leftarrow \tilde{S}_1 \cup \{a'\}$ $\quad$ {Counterfactual radius $\delta_{(0,1)}$ can be reduced, add $b$ into the training set}

22: $\quad\quad\quad$ **else**

23: $\quad\quad\quad\quad$ Repeat Line 4-25 by excluding $\delta_{(0,1)}$ from the $\max$ function $\quad$ {Counterfactual radius $\delta_{(0,1)}$ cannot be reduced, go back to other reducible covering radii from the largest one}

24: $\quad\quad\quad$ **end if**

25: $\quad\quad$ **end if**

26: $\quad\quad \mathcal{S} \leftarrow \mathcal{S} \cup \{a\}$

27: **end while**

28: **Output:** $\mathcal{S}$

---

## C. Additional Details

### C.1. Algorithm 1

**Proxy collection** $\tilde{S}_{1-t}$. To promote sample diversity in the acquired data, the reduction of the counterfactual covering radius $\delta_{(t,1-t)}$ queries the data from group $t$, however, such radius is calculated by $\delta_{(t,1-t)} = \max_{i \in \mathcal{D}_{1-t} \setminus \tilde{S}_{1-t}} \min_{j \in \mathcal{S}_t} d(\mathbf{x}_i^{1-t}, \mathbf{x}_j^t)$, where a direct acquisition from the treatment group $1 - t$ is performed. But our actual acquisition targets the query from group $t$ to reduce the counterfactual covering radius $\delta_{(t,1-t)}$ by covering the counterfactual samples. Thus, by the greedy nature of the Algorithm 1, when $\delta_{(t,1-t)}$ is the one to be reduced, the mentality for the query step is to calculate the proxy point $a' = \arg\max_{i \in \mathcal{D}_{1-t} \setminus \tilde{S}_{1-t}} \min_{j \in \mathcal{S}_{(t,1-t)}} d(\mathbf{x}_i^{1-t}, \mathbf{x}_j^t)$ from group $1 - t$, then find the nearest point $a_t \in \mathcal{D}_t$ to $a'$ as the eventual query to expand $\mathcal{S}_t$, and adding the proxy pint $a'$ into the proxy collection $\tilde{S}_{1-t}$ as a already marked position.

**Rationale of querying factual samples:** Note that querying the factual sample to reduce the counterfactual covering radius is not a necessity to help reduce the counterfactual covering radius, because a direct acquisition on the counterfactual samples can indeed reduce the counterfactual covering radius by the rigorous math definition. However, only acquiring counterfactual samples impedes the sample diversity in the counterfactual group. for example, querying the counterfactual sample fall under the neighborhood of an acquired counterfactual sample to reduce the counterfactual covering radius can cause redundancy under limited budget.

### C.2. Algorithm 2

The data acquisition performed in this algorithm is mainly repeating the two steps: 1). Pick the node with the highest scaled out-degree (which is recalculated each round for new point selection); 2). remove the incoming factual/out-going counterfactual edges to the picked node and its neighbors. Note that the scaled out-degree is by multiplying the out-degree with an coefficient $c$ that is directly associated with the covering radii and the covered points to further balance the distribution discrepancy. That is, given the union of factual and counterfactual covering ball induced by center $\mathbf{x}$: $\mathcal{A}_{(t,t)}(\mathbf{x}) \bigcup \mathcal{A}_{(t,1-t)}(\mathbf{x})$, three main possible scenarios are:

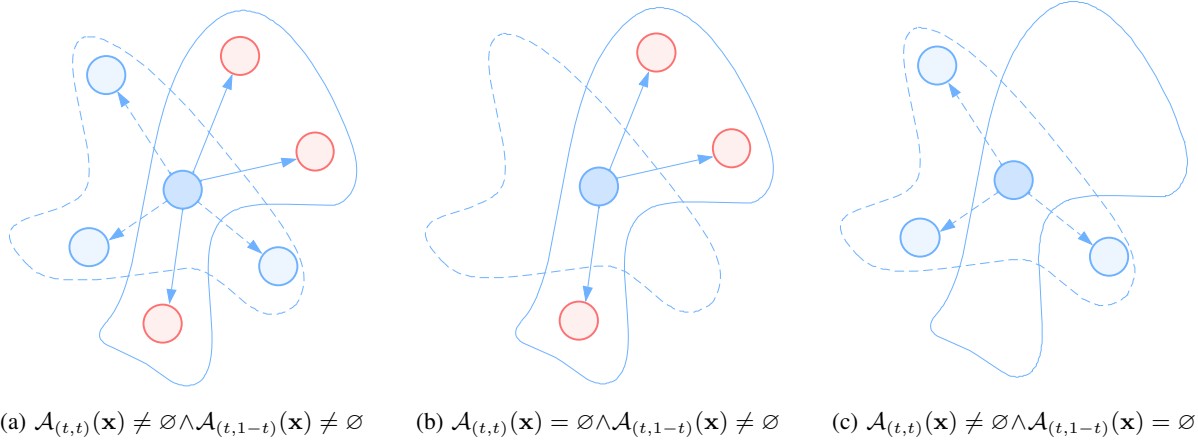

(a) $\mathcal{A}_{(t,t)}(\mathbf{x}) \neq \varnothing \wedge \mathcal{A}_{(t,1-t)}(\mathbf{x}) \neq \varnothing$      (b) $\mathcal{A}_{(t,t)}(\mathbf{x}) = \varnothing \wedge \mathcal{A}_{(t,1-t)}(\mathbf{x}) \neq \varnothing$      (c) $\mathcal{A}_{(t,t)}(\mathbf{x}) \neq \varnothing \wedge \mathcal{A}_{(t,1-t)}(\mathbf{x}) = \varnothing$

*Figure 7.* Visualization of the factual covering (dashed manifold) and the counterfactual covering (solid manifold) on the local neighborhood from the center $\mathbf{x}$. Note that the zero-neighbor scenario is omitted as it is the least preferred to maximize the coverage.

Thus, the scaling coefficient $c$ is calculated as:

$$c(\mathbf{x}) = \zeta(\mathbf{x})(1 - \zeta(\mathbf{x})), \forall \zeta(\mathbf{x}) = \frac{|\mathcal{A}_{(t,t)}(\mathbf{x})|}{|\mathcal{A}_{(t,t)}(\mathbf{x}) \bigcup \mathcal{A}_{(t,1-t)}(\mathbf{x})|} \in [0, 1], \tag{53}$$

where the minimum and maximum of the coefficient $c$ is obtained respectively by $\zeta = 0$ (Figure 7(b) and 7(c)) and $\zeta = \frac{1}{2}$ (Figure 7(a)). Noting that the shape of the coefficient $c$ w.r.t. $\zeta$ over the interval [0,1] is a downward-opening parabola, which gives the maximum $c$ on the evenly divided scenario, with $c$ declines when $\zeta$ departing from $\frac{1}{2}$. Additionally, when $c = 0$,

scenario 7(b) is preferred to scenario 7(c) from overlapping perspective with higher number of counterfactual neighbors prioritized.

## C.3. Dataset Details

**Toy:** The 2-dimensional dataset is generated by creating multiple clusters for each treatment group $t$. Let center of cluster $v_i^t = (x_{i,1}^t, x_{i,2}^t)$ to be drawn randomly from the uniform distribution:

$$x_{i,1}^t, \ x_{i,2}^t \sim \text{Uniform}([-9 + \beta_t, 9 + \beta_t]), \tag{54}$$

where the $\beta_t$ is a offset to create larger distribution discrepancy between two treatment groups. Let the collection of the centers of size $k$ for group $t$ be $\mathcal{V}_k^t = \{v_i^t\}_{i=1}^k$, the randomly added center $v_k^t$ should satisfy:

$$d_{v' \in \mathcal{V}_{k-1}^t}(v_k^t, v') \geq 1.5 \times 0.9^j, \tag{55}$$

where $j$ counts from 0 with unit increment each time when Eq. (55) is not satisfied for the randomly generated center $v_k^t$ over 100 times. That is, we reduce the minimum distance to a smaller one if the qualified sample cannot be found from the remaining pool set.

Given the centers, $\mathcal{V}_{n_t'}^t$, we define the mean $\mu_j^t = \mathcal{V}_{n_t'}^t[j]$ and variance $\sigma_j^2 = 1$ and generating $\mathbf{X}^t(i) \in \mathbb{R}^{n_t \times 2}$ for cluster $i$:

$$\mathbf{X}_j^t(i) \sim \mathcal{N}(\mu_j^t, \sigma_j^2), \ \forall j \leq n_t \implies \mathbf{X}^t = \bigcup_{i \leq n_t'} \mathbf{X}^t(i). \tag{56}$$

Thus, set $\beta_1 = 2$ and $\beta_0 = -2$, number of clusters $n_1' = 50$ and $n_0' = 30$, number of samples for each cluster $n_1 = n_0 = 200$, such that we generate covariate matrix $\mathbf{X}^{t=1} \in \mathbb{R}^{10000 \times 2}$ for treatment group $t = 1$ and covariate matrix $\mathbf{X}^{t=0} \in \mathbb{R}^{6000 \times 2}$ for treatment group $t = 1$. See visualization in main text Figure 6(a) for the generated data. Furthermore, let $\mathbf{T} \in \mathbb{R}^{(n_t' \cdot n_t) \times 1}$ be the treatment indicator, to simulate the response curve $y$, we have:

$$y_i^t = \sin(1.5 \times x_{i,1}^t) + \cos(1.5 \times x_{i,2}^t) + 5t, \forall (\mathbf{x}_i^t, t) \in (\mathbf{X}^t, \mathbf{T}). \tag{57}$$

**IBM (Shimoni et al., 2018):** This dataset is based on the real-world 177 covariates from a cohort of 100,000 individuals, from the publicly available Linked Births and Infant Deaths Database. The generated response curve is based on the randomly selected 25,000 individuals out of the 100,000 base, and the potential outcomes have 10 different simulations according to (Shimoni et al., 2018).

**CMNIST (Jesson et al., 2021a):** This dataset contains 60,000 image samples (10 classes) of size 28×28, which are adapted from MINIST (LeCun, 1998) benchmark. CMNIST is completely distinct from the previous tabular datasets by leveraging the image data for the treatment effect estimation. The potential outcomes are simulated 10 times and generated by projecting the digits into a 1-dimensional latent manifold as described in (Jesson et al., 2021a).

## C.4. Further Discussions

It is seen that in Section 5.3 for ablation study, the performance difference shown in Table 2 indicates that the no measurable and significant performance gain on IBM and CMNIST dataset, respectively. Here, we qualitatively discuss the causes for this phenomenon from the data distribution perspective.

The entire data distribution for IBM is plot in Figure 8(a), where a well-blended distribution is seen. We further split the entire data distribution into sole plots on Figure 8(b) ($t = 1$) and Figure 8(c) ($t = 0$). It is observed that data distributions on different treatment groups are almost identical, which is aligned to the dataset description in the original paper that generates the IBM benchmark (Shimoni et al., 2018). Thus, the high-density region is built on the overlapping region since two groups are fully overlapped. Therefore, given all the samples on group $t = 0$ known, when data acquisition starts on group $t = 1$, algorithm only needs to acquire the sample from the high-density regions as the overlapping condition is synchronously satisfied. Thus, FCCM and FCCM$^-$ is indistinguishable on such data distribution. In the meanwhile, it is observed that the entire CMNIST data distribution (Figure 8(d)) embeds significantly less overlapping regions with the covariates extracted from the MNIST (LeCun, 1998) benchmark which has 10 classes of data (10 digits). Unlike IBM (Shimoni et al., 2018), the data generating process in (Jesson et al., 2021a) creates significant distribution discrepancy for two treatment groups, where

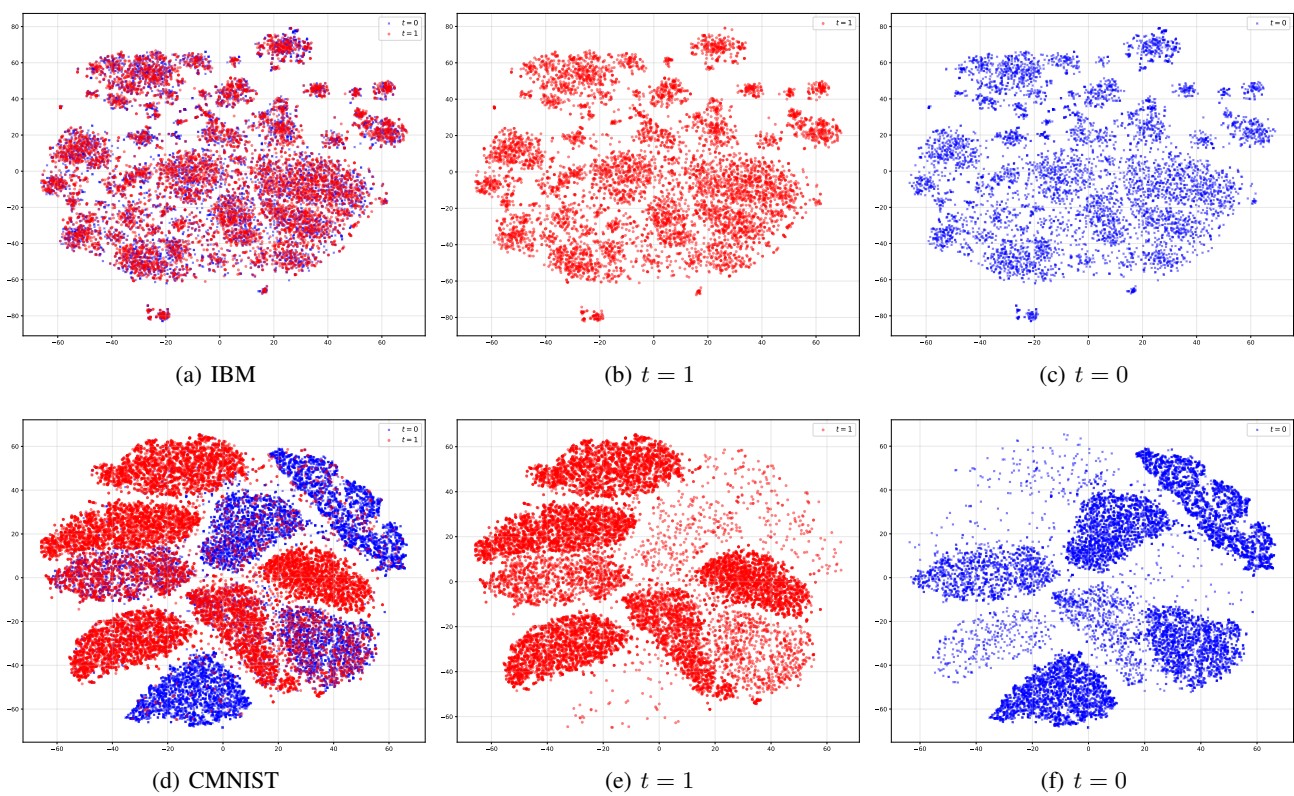

*Figure 8.* High dimensional data projected into 2-dimensional space via t-SNE, with colors indicating overlapping, group $t = 1$, and group $t = 0$. Left: entire data distribution; Mid: distribution on group $t = 1$; Right: distribution on group $t = 0$.

the high-density regions for group $t = 1$ (Figure 8(e)) and group $t = 0$ (Figure 8(f)) are well-distinguished, leaving less overlapping regions on the entire data distribution. Therefore, our proposed method – FCCM prioritizes the data acquisition toward the high-density and overlapping region, while FCCM$^-$ only focus on high-density region where less overlapping is seen, thus resulting in significant performance difference as seen in Table 2.

### C.5. Practicability of the Assumptions

Strong ignorability: The validity of the SI can be approximated by carefully selecting sufficient relevant covariates and constructing a more balanced dataset.

Lipschitz continuity: Theorem 3.3 assumes the Lipschitzness of the $p^t(y|\mathbf{x})$, this can be a practical assumption if the regression model $f$, e.g., a well-regularized neural network (NN), learns a smooth mapping from $\mathbf{x}$ to $y$, which implies the Lipschitzness for $p^t(y|\mathbf{x})$. Also, the NN $f$ is differentiable w.r.t. $\mathbf{x}$, thus the squared loss $l_f$ is also bounded and sufficiently differentiable w.r.t. $\mathbf{x}$, further implying the Lipschitzness of $l_f$.

Constant $\kappa$: The existence of the constant $\kappa$ is thoroughly discussed by Shalit et al. (2017) in Appendix A.3 and A.4.

Additionally, we discuss consequence for the proposed theorems when the above-mentioned assumptions are not satisfied:

Lipschitz continuity for Theorem 3.4: If the Lipschitzness does not hold, the multiplicative constant will be unbounded, and thus the reduction of the radii may not help control the risk upper bound. Strong ignorability for Theorem 4.1: Strong ignorability provides an ideal scenario for acquiring the counterfactual samples, where a quick bound reduction by Algorithm 1 is seen in Figure 2(a). If strong ignorability cannot be guaranteed in real-world data, e.g., CMNIST, the reduction of the bound will be significantly slower and less effective for Algorithm 1, as shown in Figure 2(b). Thus, it motivates us to propose Algorithm 2 that can handle compromised data distributions more effectively via a slight trade-off on coverage.

## C.6. Hyperparameters Tuning

We conduct all the experiments with 24GB NVIDIA RTX-3090 GPU on Ubuntu 22.04 LTS platform with the 12th Gen Intel i7-12700K 12-Core 20-Thread CPU. As stated in the main text, for fair comparison, we take the consistent hyperparameters tuned in (Jesson et al., 2021b; Wen et al., 2025) for the estimators: **DUE-DNN** (Van Amersfoort et al., 2021) and **DUE-CNN** (Van Amersfoort et al., 2021) shown in Table 3. Additionally, we search the best hyperparameters, i.e., covering radius $\delta$ and edge weight $\alpha$ for counterfactual linkage, for Algorithm 2 with the validation set shown in Table 4. Note, that in Section 4.3, we approximating a narrower range around 95% threshold to further determine the covering radius for the best performance.

*Table 3.* Hyperparameters for Estimators

| Hyperparameters | DNN | CNN |
|---|---|---|
| Kernel | RBF | Matern |
| Inducing Points | 100 | 100 |
| Hidden Neurons | 200 | 200 |
| Depth | 3 | 2 |
| Dropout Rate | 0.1 | 0.05 |
| Spectral Norm | 0.95 | 3.0 |
| Learning Rate | 1e-3 | 1e-3 |

*Table 4.* Hyperparameters for Algorithm 2

| Hyperparameters | Search Space | Tuned |
|---|---|---|
| $\delta_{(1,1)}$ for TOY | [0.11, 0.12, 0.13] | 0.11 |
| $\delta_{(1,0)}$ for TOY | [0.11, 0.12, 0.13] | 0.11 |
| $\delta_{(1,1)}$ for IBM | [0.11, 0.13, 0.15] | 0.11 |
| $\delta_{(1,0)}$ for IBM | [0.11, 0.13, 0.15] | 0.11 |
| $\delta_{(1,1)}$ for CMNIST | [0.40, 0.45, 0.50] | 0.50 |
| $\delta_{(1,0)}$ for CMNIST | [0.40, 0.45, 0.50] | 0.40 |
| Edge weight $\alpha$ | [1.0, 2.5, 5.0] | 2.5 |

## C.7. Main Results under Highest Resolution

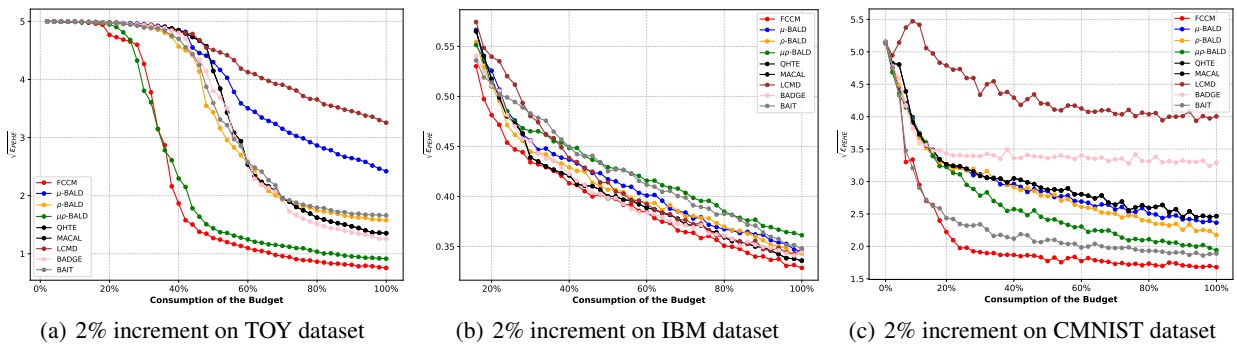

(a) 2% increment on TOY dataset    (b) 2% increment on IBM dataset    (c) 2% increment on CMNIST dataset

## C.8. Sensitivity Study

Note that the acquisition on treatment sample $t = 1$ is insensitive on $\delta_{(0,0)}$ and $\delta_{(0,1)}$ in our setting, as all control samples ($t = 0$) are seen. For $\alpha$, our setting of $\alpha = 2.5$ has an overall lower error across different acquisition budgets.

(d) Analysis for the weight $\alpha$ on TOY    (e) Analysis for the weight $\alpha$ on IBM    (f) Analysis for the weight $\alpha$ on CMNIST

(g) Analysis for $\delta_{(1,1)}$ on TOY    (h) Analysis for $\delta_{(1,1)}$ on IBM    (i) Analysis for $\delta_{(1,1)}$ on CMNIST

(j) Analysis for $\delta_{(1,0)}$ on TOY    (k) Analysis for $\delta_{(1,0)}$ on IBM    (l) Analysis for $\delta_{(1,0)}$ on CMNIST

