# OpenReview forum: "Enhancing Treatment Effect Estimation via Active Learning: A Counterfactual Covering Perspective"
_ICML.cc/2025/Conference — ICML 2025 poster_

### Official Review · Reviewer_CkNS · 2025-03-09

**Overall Recommendation:** 2

**Summary:**

This paper proposes a data-efficient method to construct a regression model for estimating the mean difference between the treatment and control responses. Specifically, the author upper-bounds the evaluation error and proposes two radius-based approaches to minimize the upper bound with a limited label budget.

**Claims And Evidence:**

My primary concern regarding the paper is that the proposed approaches bear a stronger resemblance to coreset learning rather than active learning. In active learning, users interact with the real world, acquire responses (i.e., labels), and utilize these responses to refine the learning process. However, as seen in Algorithm 1 and Eq. (8), the core of the proposed approach focuses on finding a coreset that covers the samples in the treatment and control groups. In other words, there is no interactive learning in which the treatment/control responses are contributory.

**Essential References Not Discussed:**

I am not aware of.

**Experimental Designs Or Analyses:**

I have a bare-minimum check on the experimental results, as the methodology part raises significant concerns.

**Methods And Evaluation Criteria:**

The authors evaluate the performance of the proposed algorithm across various label budgets and compare them with multiple baseline. This makes sense.

**Other Comments Or Suggestions:**

In assumption 2.3, should use $T$ to indicate a random variable and and use $t$ to indicate the realization., e.g., $p(Y=1\mid x)$.

**Other Strengths And Weaknesses:**

I think that lines 6 and 7 in Algorithm 1, which involve searching for points, are critical. However, the authors have included these parts in the appendix, and it seems they did not provide the intuition behind these queries in the main paper. It would be helpful to include an explanation of what data should be selected to help reduce the maximum radius.

**Questions For Authors:**

Please see the Theoretical Claims.

**Relation To Broader Scientific Literature:**

The paper states that the proposed method can be applied to evaluate the difference between treatment and control responses in a label-efficient manner.

**Theoretical Claims:**

I did not check the correctness of proofs. However, I have a couple of concerns regarding the theory.\
(1) \mathbf{Theorem 3.3}: What is $\mathcal{R}$? What is the hypothesis class $H$ comprised of? Does $H$ contain regression functions which are used to model among $x,t$ and $y^t$?\
(2) Corollary 3.4: Shouldn’t this be a $(1-\gamma)$ probability-type result? I am thinking the response $y$ is noisy and there is chance even $\gamma$ is small, $\Delta$ could also be big.

---

> ### Author Rebuttal · Authors · 2025-03-31
>
> **Active Learning vs. Coreset Learning:** The problem setting we described in lines 114-127 (left column) is an active learning loop that consists of 5 steps, where in Step 4 we identify a subset from the completely unlabeled pool set $\mathcal{D}=\{(\mathbf{x}_ {i},t_ {i})\}^{N}_ {i=1}$ for the oracle to obtain the labels (i.e., treatment outcome $y_i$). The newly labeled samples are then used to refine the estimation model in Step 5. As such, our setting differs from coreset learning that operates with all samples' labels $y_i$. As our labels are actively acquired, Algorithm 1 corresponds to Step 4 (unlabeled subset identification from $\mathcal{D}$) of our problem setting. In Eq.(8), though it is formulated using the labeled training sets $\mathcal{S}_ {0}$ and $\mathcal{S}_ {1}$, the calculation of Eq.(8) only requires covariate $\mathbf{x}_i$ and its corresponding treatment assignment $t_i$ (lines 202-203 right column), hence are applicable to all samples in the unlabeled pool set $\mathcal{D}$. In both Eq.(8) and Algorithm 1, we slightly abused the notation $\mathcal{S}$ to make the notations less cluttered, as their calculations do not actually rely on the access to treatment outcome labels $y_i$. To avoid potential confusion, we will update the description and notation in corresponding parts to highlight the active learning setting.
>
> **$\mathcal{R}$, and Class $\mathcal{H}$:** Thanks for pointing out the typo "$\mathcal{R}$'', we will correct it to the feature space "$\mathcal{X}$''. The family of functions $\mathcal{H}$={$h \mid h: \mathcal{X} \rightarrow \mathbb{R}$} does contain the regression model $f$ as part of the function $h$. As per Theorem 3.3, we have $h_{f}(\mathbf{x},t):=\frac{1}{\kappa}l_{f}(\mathbf{x},t)\in \mathcal{H}$ ($t$ is a constant for any given $\mathbf{x}$), which contains the loss function $l_{f}$ to evaluate regression model $f$. **The rationale for defining $\mathcal{H}$ is that** the constant $\kappa_{\mathcal{H}}$ in our derived bound is dependent on $\mathcal{H}$. As per line 966, the term in Eq.(32c) is further upper-bounded by Integral Probability Metric (IPM) under the premise that $h_{f}(\mathbf{x},t)$ is in $\mathcal{H}$, which is a useful bounding technique also seen in (Shalit et al., 2017).
>
> **Corollary 3.4 and Noisy Response $y$:** Thanks for improving the completeness of the Corollary, which should be claimed with the probability of least $1-\gamma$ since it is a further simplification of Theorem 3.3. Its main idea is to reveal what optimizable terms are closely related to risk convergence. For the gap $\Delta$, we follow the common practice in (Sener \& Savarese, 2018) by assuming zero training loss and noiseless observation for the theoretical analysis purpose. Under the active learning setting, it is reasonable that the labels obtained from the oracle are sufficiently reliable, i.e., $y_i$ has minimal noise.
>
> **How We Query in Algorithm 1:** As stated in lines 215-219 (right column), the intuition behind the query process is to greedily prioritize the reduction of the largest radius, which is a commonly used intuition also seen in (Tsang et al., 2005; Sener \& Savarese, 2018) for the $k$-Center problem. As per Eq.(8), each of the four radii is defined in lines 202-204 (right column), thus, we just need to query the point (no label) that constitutes the radius $\delta$, and the query is explicitly defined in line 6 of the full Algorithm 1 (Appendix C) if we need to reduce $\delta_{(1,1)}$. The visualizations in Figure 1 are a graphical expression of Algorithm 1's intuition, i.e., the point at the other end of the radius is the next acquisition goal. Notably, Algorithm 1 is a naive instantiation of the covering radius-based active learning and comes with a strong (yet impractical) assumption on fully overlapping data distributions (lines 232-234 right column). Thus, we made its description relatively brief to pave the way for our proposed main algorithm FCCM (Algorithm 2), which relaxes the full coverage constraint to achieve stronger radius reduction under the same labeling budget. We will add this explanation when introducing Algorithm 1 to provide more intuitions.
>
> **Random Variable $T$:** Thanks for pointing out the typo, we will correct $t$ to $T$ where needed.

---

> > ### Comment · Reviewer_CkNS · 2025-04-04
> >
> > Thank you for your response. I will withhold my judgment on the score for now, primarily because the labels $y$ have not been incorporated into the acquisition function in Eq. (8). While selecting informative points based solely on covariates and treatment and response assignment may provide improvement over the random querying, it is far less effective than actively involving the labels $y$ in the decision process. In fact, a key advantage of active learning lies in its ability to avoid querying points that are unlikely to provide additional value based on the observed labels. I think important consideration is missing from the proposed approach

---

> > > ### Author Response · Authors · 2025-04-05
> > >
> > > We greatly appreciate reviewer CkNS's responsive reply and valuable time during the discussion phase. We address the raised concerns as follows:
> > >
> > > **Unlabeled data selection with existing labels $y_{i}$ in the training set:** We would firstly address this concern by distinguishing the difference between the **Uncertainty-based Querying** and the **Diversity-based Querying**. As in the uncertainty-based querying, e.g., BALD (Gal et al., 2017), BAIT (Ash et al., 2021), Causal-BALD (Jesson et al., 2021b), the label $y_{i}$ is used for querying further unlabeled samples because the proposed methods are **model-dependent**, which requires training an uncertainty-aware model based on the existing labeled training set, thus further querying on the unlabeled sample involves the uncertainty output from the trained model. Also, the model-dependent method requires accurate uncertainty estimation, please refer to **Q1** for reviewer **hAfw** for elaborating on the drawback in model-dependent methods. **However**, the diversity-based querying is **model-independent**, e.g., Core-Set (Sener \& Savarese, 2018), ProbCover (Yehuda et al., 2022), MACAL (Wen et al., 2024), which queries the unlabeled samples from the diverse regions of the data distribution purely based on the covariates (and treatment if for causal inference) and Eq. (8) is a classical form of the core-set querying without the labels. **Note,** that our task focuses on the regression as stated in line 025 (right column), where labels are distributed in the continuous space, such that each unique sample $\mathbf{x}_{i}$ is assigned with a unique label $y_i$, so the diversity-based querying on the feature space natural guarantees the diversity in the label space.
> > > **Why Diversity-based querying in our paper?** We would like further to address the method of choice in our paper. **It is noted that our paper is theory-based**. Our main theorem -- Theorem 3.3 suggests that the incalculable estimation risk PEHE is upper-bounded by four computable radii, **which unveils the independence from the label information under our theoretical framework and the proposed theorem deeply motivates our algorithm designed to reduce the covering radii**. Thus, to reduce the key terms in the risk upper bound, we first propose a straightforward radius reduction method in Algorithm 1. By considering the limitation of Algorithm 1, we then further propose Algorithm 2 to accelerate the radius reduction specifically tailored for the treatment effect estimation setting. In summary, incorporating the label information for querying depends on the choice of the method, and most importantly, our proposed interesting and intuitive theorem for the treatment effect estimation problem with active learning unveils the independence of the label information in the risk upper bound.
> > >
> > > We will incorporate the discussion to update our paper regarding the use of the label information and we kindly welcome any further concerns from reviewer CkNS.

---

### Official Review · Reviewer_FYYH · 2025-03-10

**Overall Recommendation:** 4

**Summary:**

This paper tackles causal effect estimation in active learning, where only features and treatments are known, but outcome labeling is costly and incomplete. To ensure balanced label expansion, it bounds generalization error using factual and counterfactual covering radius. Starting with a greedy approach, it develops the FCCM algorithm, optimizing coverage in real-world data. Experiments on benchmarks show state-of-the-art performance in causal effect estimation.

**Claims And Evidence:**

The claims are supported by the evidence.

**Essential References Not Discussed:**

N/A

**Experimental Designs Or Analyses:**

The experimental results show a promising level of performance gain from the proposed FCCM algorithm. The settings are rational and the radius hyperparameters are provided, where the algorithm is made available for reproducibility. On top of the quantitative results, I find the visualizations in Figure 6 also helpful for showing FCCM’s advantages.

**Methods And Evaluation Criteria:**

The method is developed by extending the covering radius concept to the counterfactual sample setting, which in my view is a neat and intuitive solution to the problem. The evaluation adapts major benchmark datasets in causal effect estimation to the active learning setting, and covers a variety of strong baselines. Thus, the methodology and experiment sections are rigorous.

**Other Comments Or Suggestions:**

Typos/presentation issues:
1 Line 380, “a fine granularity results” should be “fine-granularity results”
2 Line 845 in Appendix, “distributions” is mis-spelled
3 Line 957 in Appendix, the meaning of “by change of variable reverting y to x” is unclear

**Other Strengths And Weaknesses:**

Strengths

1 This paper clearly differentiates the problem it studies with another line of work in data-efficient causal inference, which is active experiment design. The practical scenarios where only the treatment effect (but not treatment) is unknown is well motivated and presents some important challenges unique to the problem.

2 The solution approaches treatment-aware data selection/labeling from an inter- and intra-group covering radius perspective, which is a technically sound and innovative solution.



Weakness

1 In terms of the experimental setting, as each dataset has 50 acquisition steps, it might help demonstrate the performance trend of FCCM if the results at more budget levels can be shown. For example, a figure similar to Figure 5 can be added to Appendix to showcase the prediction performance with a finer-grained increment (e.g., 10% or even 2% percent).

2 The presentation of the ablation study results in Table 2 is somewhat untraditional. It would be more intuitive to directly report the PEHE values of both variants, with the calculated percentage difference as an additional reference.

3 The active learning strategy needs to maximize a synergic objective with four covering radii (two inter-group and two intra-group) combined, where the authors state that all four radii share the same setting. It is worth discussing whether or not using a uniform hyperparameter setting for all four radii is the best possible practice.

4 In Figure 5, the reported results for both TOY and CMNIST datasets start from 0% budget, while IBM starts from 20%. Is it because the error at zero budget is too large for IBM? Since the causal estimator is the same across all active learning methods, the results at 0% are in fact less informative and can be removed for TOY and CMNIST.

**Questions For Authors:**

1 The Introduction mentions a real-world application for “collecting customer preferences from those who have already received different services”. The applicability of the studied problem to this example is not very clear to me; can you explain with more details?
2 How does FCCM’s performance look like when using a finer-grained increment of the labeling budget?
3 Any insights into sharing the same hyperparameter value among all four radii? In other words, in what scenarios will different radii values help?
4 Is there a reason that results for IBM with 0% budget are omitted?
5 Can the authors provide the PEHE results of FCCM and FCCM- from the ablation study?

**Relation To Broader Scientific Literature:**

This paper resolves the issue of limited labeled data in the causal inference space, which aligns with real-world challenges when building data-driven models under data scarcity due to cost and/or privacy concerns. Hence, the relevance to the broader of causal inference is high.

**Theoretical Claims:**

(1) In the proof of Theorem 3.3, it would be useful to explicitly describe the magnitude of $\lambda$, such that the probability of $1-\lambda$ is sufficiently large to be practical.

(2) In the proof of Lemma A.3, the acronym SUTVA should be expanded as it is not introduced elsewhere.

---

> ### Author Rebuttal · Authors · 2025-03-31
>
> **Q1:** For instance, when performing A/B tests to compare two software versions, the service provider needs to assign two distinct user groups to different versions (i.e., treatments). To understand which software version offers a better experience for a specific user profile (i.e., features), it is necessary to collect users' feedback via customer surveys. Though the features and treatments associated with each user are known to the service provider, collecting all treatment outcomes -- users' satisfaction levels is time-consuming and economically unrealistic (e.g., paid surveys). In this case, active learning can be used to select a small subset of the most informative users, from whom the treatment outcomes can be obtained at a much lower cost.
>
> **Q2+W1:** We provide the figures under the 2\% increment evaluation scheme, which is the most fine-grained setting based on the 50 query steps in total in the anonymous link https://anonymous.4open.science/r/To-Reviewer-FYYH (click into folder ``To-Reviewer-FYYH''). Note that on the IBM dataset, all methods yield an impractically large prediction error at the initial acquisition steps (<20\% budget), thus we show the results from 20\% for better clarity.
>
> **Q3+W3:** Our main rationale is to avoid making the search space prohibitively large -- up to $\mathcal{O}(n^4)$ if each radius has $n$ candidate settings. The key insight to reduce the search complexity is that, because $P(\mathcal{A})=\frac{1}{4}(P(\mathcal{A}^{t=1}_ {F})+P(\mathcal{A}^{t=0}_ {F})+P(\mathcal{A}^{t=1}_ {CF})+P(\mathcal{A}^{t=0}_ {CF}))$, each radius is **independent** and each of the sub-terms increases monotonically with the radius, then so does the mean coverage $P(\mathcal{A})$. Thus, by the independence and monotonicity, the search space greatly reduces to $\mathcal{O}(n)$ while the search of four radii stays in the same direction, thus we let the four radii share the same setting. **When will different radii help?** Though the shared value among four radii allows for effective and efficient hyperparameter tuning, it could potentially fail if the distribution discrepancy between the two treatment groups is very large. This is because the counterfactual covering radius can be $\delta_{(t,1-t)}\gg\delta_{(t,t)}$ for the counterfactual coverage $P(\mathcal{A}^{t}_{CF})$ to get close to full, lowering the utility of the uniform radii value. As such, if the distribution discrepancy between treatment groups is large, we can then identify a different value for each covering radius to maintain the high coverage under the linear complexity due to the independence and monotonicity. We will add this discussion to an updated version of our paper.
>
> **Q4+W4:** The reason for omitting 0\% in IBM is that the estimation risk at the first few steps is too large to be practical for all methods. Thus, the reported results start from 20\% for clarity. We will append a more explicit explanation on the figure caption to eliminate potential confusion.
>
> **Q5+W2:** Thanks for the suggestion, we will update the table to show the PEHE results for both FCCM and FCCM$^-$ in our paper. Please check the anonymous link https://anonymous.4open.science/r/To-Reviewer-FYYH (click into folder ``To-Reviewer-FYYH'') for the detailed results.
>
> **Probability Threshold $\lambda$ ($\gamma$ in Our Paper):** We will add a discussion of the magnitude of the probability threshold $\gamma$. It is noted that higher confidence gives a larger upper bound due to the increase in the third term of the bound, but it remains a constant during our optimization.
>
> **SUTVA:** We will update it with the full spell, namely Stable Unit Treatment Value Assumption (SUTVA).
>
> **Typos:** Thanks for pointing them out, and we will correct them correspondingly. For ``by change of variable reverting y to x'', we intended to express that the integral over the domain $\mathcal{Y}$ is changed to the domain $\mathcal{X}$ by the change of variable. We will modify the description to clear the confusion.

---

### Official Review · Reviewer_hAfw · 2025-03-10

**Overall Recommendation:** 3

**Summary:**

This paper proposes an active learning (AL) approach tailored to enhance the treatment effect estimation from observational data, when labeling outcomes is costly.  The authors introduce a theoretical formulation using the concepts of factual/counterfactual covering radii, to upper bound the (fundamentally incalculable) population-level risk --- expected precision in estimation of heterogeneous effect (PEHE), defined in Eq. (3) --- effectively by the sum of four factual/counterfactual covering radii, up to a multiplicative constant and lower-order terms, cf. Eq. (4) and Corollary 3.4.

Two distinct greedy algorithms to minimize the covering radii are introduced: (1) A direct radius-reduction algorithm (Algorithm 1 in Section 4.1) motivated by traditional core-set techniques, and (2) the factual and counterfactual coverage maximization (FCCM) method (Algorithm 2 in Section 4.2), which seeks a more balanced and effective radius reduction through maximizing coverage under given radii constraints. Empirical experiments conducted on synthetic, semi-synthetic, and image-based datasets demonstrate that the FCCM approach notably outperforms established baselines by systematically prioritizing samples from regions of high density and significant overlap between treatment groups.

**Claims And Evidence:**

The main claims---(1) a counterfactual covering perspective provides a tractable risk upper bound, and (2) the proposed FCCM enhances estimation performance and improves theoretical risk upper bounds---are well-supported by empirical evidence provided through experiments. The choice of datasets (TOY, IBM, and CMNIST) effectively demonstrates the general applicability and performance advantages of the FCCM approach. Nonetheless, a deeper discussion linking theoretical guarantees to empirical outcomes, especially considering variations in dataset properties such as the degree of overlap between treatment groups, would enhance the validity of these claims. While the empirical evidence is strong, a more explicit acknowledgment of potential failure modes or limitations of the proposed methods could strengthen the authors’ analysis and presentation.

**Essential References Not Discussed:**

The discussion of related literature provided in Appendix B seems mostly sufficient.

**Experimental Designs Or Analyses:**

The experimental designs are sound, leveraging suitable baselines including QHTE, Causal-BALD, MACAL, LCMD, BADGE, and BAIT, effectively situating the proposed method within existing literature. Nevertheless, the robustness of the FCCM algorithm could be more comprehensively assessed through sensitivity analyses, particularly concerning hyperparameters such as the radii parameter $\delta_{t,t'}$ (for $t, t' \in \{0,1\}$) and the weight parameter $\alpha$.

**Methods And Evaluation Criteria:**

The chosen methods and evaluation criteria---e.g., the PEHE metric---are appropriate and commonly accepted within the causal inference community. The diversity in datasets (synthetic, semi-synthetic, and image-based) effectively demonstrates methodological versatility. However, more explicit explanations and motivations for the chosen datasets in the main text would improve clarity. While Appendix C.2 provides some context, essential details such as dataset characteristics (e.g., types and ranges of covariantes and response variables) and preprocessing steps remain insufficiently clear.

**Other Comments Or Suggestions:**

The paper presents an interesting idea and a decent contribution; however, I believe its readability can be significantly enhanced.  Below I list some suggestions.

* Definitions could be streamlined to improve clarity and reduce redundancy.  For example, Definitions 3.1 and 3.2 could be merged to define covering radius $\delta_{t,t'}$ generically. which can then be specialized to define factual/counterfactual radii. The same comment also applies to Definitions 4.2 and 4.3

* The statement of Theorem 3.3 is currently dense and challenging to follow. Explicitly separating and clearly stating assumptions outside the theorem (e.g., by defining "Assumption" environment) would significantly aid comprehension. I would also suggest the authors further discuss the practical implications and limitations of the theoretical assumptions explicitly.

* The description of Section 4.2 is unclear and confusing.  I would suggest the authors revise description more precisely and compactly, e.g., by defining quantities and sets, e.g., $\delta_{\mathrm{sum}}$, $\mathcal{A}$, $P(\mathcal{A})$ explicitly using mathematical expressions instead of vaguely defining them in English.

* Sentences throughout the manuscript often tend to be overly long and verbose. Breaking these into shorter, clearer sentences would enhance readability and improve interpretability.

* Additional Minor comments
  - Please consider revising the captions more informatively. For example: (1) please add "by Algorithm 1" in Figure 2; (2) I think the caption of Figure 3 should contain "Algorithm 2" instead of "Algorithm 1"; and (3) please add "by Algorithm 2" in Figure 4.
  - I think Algorithm 2 is missing an input parameter $\alpha$.

**Other Strengths And Weaknesses:**

* Strength: The theoretical framework based on factual and counterfactual covering radii is innovative and potentially valuable for future methodological development.

* Weakness: The paper's presentation could be significantly improved to enhance clarity and readability, particularly in defining key concepts and motivating algorithmic choices.

**Questions For Authors:**

1. Could you elaborate on the following sentence you wrote in Section 5.1: "For example, although $\mu\rho$BALD bias the data acquisition toward the overlapping region, it does not further embed the property to query from the high-density region, and thus accounts for less risk" and explain the mechanism how FCCM specifically addresses distributional discrepancies compared to existing methods like $\mu\rho$BALD, particularly in scenarios with significant non-overlap? I am asking this question because (1) Figure 6-(c) seems to suggest that $\mu\rho$BALD indeed places some points outside the overlapping region, instead of "biasing the data acquisition toward the overlapping region," and (2) I want to gain intuitive understanding on how FCCM (Algorithm 2) addresses the challenge.

2. How sensitive is FCCM to the choice of hyperparameters, particularly the radii parameter $\delta_{t,t'}$'s  and the weight parameter $\alpha$? More details or experiments here would significantly enhance confidence in the robustness of your method.

3. Can you discuss the implications on your theoretical guarantees and empirical results if key assumptions, such as strong ignorability and Lipschitz continuity, do not hold in practice?

**Relation To Broader Scientific Literature:**

The paper situates its contributions within the broader literature, linking clearly to existing AL methods such as BALD and core-set approaches, as well as contemporary causal inference methods like QHTE and MACAL.

**Theoretical Claims:**

I reviewed the main theoretical claim (Theorem 3.3) and associated lemmas (A.2, A.3, and A.5). While I did not verify every line in the proofs, the theoretical derivations appear generally sound, utilizing standard techniques. However, assumptions such as strong ignorability and Lipschitz continuity, though commonly used in the literature, deserve more explicit justification regarding their practicality. Moreover, clarifying and simplifying the presentation of assumptions and key theorem statements would significantly improve readability and comprehension.

---

> ### Author Rebuttal · Authors · 2025-03-31
>
> **Potential Failure Modes:** FCCM is designed to better handle the partially overlapped data for a quicker bound reduction while maintaining high coverage (Figure 3). As such, in scenarios where the two treatment groups have no overlapping regions (e.g., biased treatment assignment), the data acquisition of FCCM will be challenged as there are no overlapping, counterfactual pairs to be identified. One possible remedy is to operate FCCM in the latent space where the inter-group distributions are aligned by methods like (Zhang et al., 2020; Wang et al., 2024), but it also offsets the model-independent advantage of FCCM. We will add this discussion in an updated version.
>
> **Practicality of Strong Ignorablity (SI):** It is noted that the focus of this paper is active label acquisition, it can then integrate with the selection bias-aware method to make predictions. The validity of the SI can be approximated by carefully selecting sufficient relevant covariates and constructing a more balanced dataset.
>
> **Practicality of Lipschitz Continuity:** Theorem 3.3 assumes the Lipschitzness of the $p^{t}(y|\mathbf{x})$, this can be a practical assumption if the regression model $f$, e.g., a well-regularized neural network (NN), learns a smooth mapping from $\mathbf{x}$ to $y$, which implies the Lipschitzness for $p^{t}(y|\mathbf{x})$. Also, the NN $f$ is differentiable w.r.t. $\mathbf{x}$, thus the squared loss $l_{f}$ is also bounded and sufficiently differentiable w.r.t. $\mathbf{x}$, further implying the Lipschitzness of $l_{f}$.
>
> **Q1:** **Elaboration on The Sentence:** Despite that $\mu\rho$-BALD ($\mu\rho$) highlights searching for samples within the non-overlapping region, its acquisition criterion naturally prefers samples that lead to higher estimated uncertainty by its criterion. **However**, the criterion design of $\mu\rho$ is not density-aware. For example, in early acquisition steps, two unseen data points $a$ and $b$ can be of the same uncertainty, while $a$ is an outlier yet data $b$ belongs to a dense cluster (not seen). Then, prioritizing the acquisition of $b$ over $a$ can help generalize the estimator to more samples, which means training on $b$ can eliminate more estimation risk when data is limited. However, $\mu\rho$ does not differentiate the importance of $a$ and $b$, and thus "account for less risk'' in such cases. **How FCCM addresses this challenge:** Pertaining to the previous example, the factual covering from FCCM can prioritize the acquisition of $b$ over $a$, as $b$ has more neighbors (i.e., higher out-degree) than $a$ in the graph constructed in Algorithm 2. To enhance the distribution alignment, FCCM makes use of the weight $\alpha$ imposed on the counterfactual edges (line 308 left column). For example, assume $c$ is in the same treatment group as $b$, while $b$ and $c$ have similar numbers of neighbors in the graph. Then, if $c$ has more neighbors from the counterfactual group than $b$ does, then the priority of $c$ is higher than $b$ owing to the additional bonus (with $\alpha>1$) on counterfactual edges when calculating the out-degree. For cases with significant non-overlap (almost no overlap), please refer to our response to potential failure modes. **Why $\mu\rho$ queries non-overlapping samples in Figure 6(c):** As stated in line 85, $\mu\rho$ relies heavily on the accuracy of estimated uncertainty (model-dependent), which can be erroneously high for certain samples outside the overlapping region, boosting the acquisition metric score. Thus, a few such samples are selected as per Figure 6(c). In contrast, FCCM is model-independent, and the acquisition score is robustly computed via the out-degree of nodes (Algorithm 2).
>
> **Q2:** Please see the anonymous link: https://anonymous.4open.science/r/To-Reviewer-hAfw for sensitivity analysis results on $\delta$ and $\alpha$ (click into folder "To-Reviewer-hAfw''). Note that the acquisition on treatment sample $t=1$ is insensitive on $\delta_{(0,0)}$ and $\delta_{(0,1)}$ in our setting, as all control samples ($t=0$) are seen. For $\alpha$, our setting of $\alpha=2.5$ has an overall lower error across different acquisition budgets. We will include all results in an updated version.
>
> **Q3:** **Theorem 3.3 (Lipschitz continuity):** If the Lipschitzness does not hold, the multiplicative constant will be unbounded, and thus the reduction of the radii may not help control the risk upper bound. **Theorem 4.1 (SI):** SI provides an ideal scenario for acquiring the counterfactual samples, where a quick bound reduction by Algorithm 1 is seen in Figure 2(a). If it cannot be guaranteed in real-world data, e.g., CMNIST, the reduction of the bound will be significantly slower and less effective for Algorithm 1, as shown in Figure 2(b). Thus, it motivates us to propose Algorithm 2 which can handle compromised data distributions more effectively via a slight trade-off in coverage.
>
> **We will adopt all the constructive suggestions to update the paper.**

---

> > ### Comment · Reviewer_hAfw · 2025-04-04
> >
> > Thank you for the response.  With the proviso that the authors will incorporate appropriate revisions to clarify the discussed aspects, I have increased my evaluation rating from 2 to 3.

---

> > > ### Author Response · Authors · 2025-04-05
> > >
> > > We again deeply appreciate Reviewer hAfw’s valuable time and constructive suggestions, which have been instrumental in helping us further improve the paper. We will update the paper accordingly.

---

### Official Review · Reviewer_8nJA · 2025-03-13

**Overall Recommendation:** 4

**Summary:**

This paper tackles the problem of active learning for treatment effect estimation, where the task is to label a treated or untreated unit given a fixed budget. The authors take the core set approach and extend it by introducing a counterfactual covering radius along with the factual covering radius. An algorithm FCCM is then introduced that chooses to label data points to minimises the expected factual and counterfactual coverings. Through quantitive experiments, the authors show that their method reduces the PEHE metric faster that other methods. Qualitatively, the authors also show that their method is better at choosing data points from the overlap of treatment groups, which is more informative of the treatment effect.

## update after rebuttal
I thank the authors for their response, I will keep my already positive score.

**Claims And Evidence:**

The claims seem to be supported as their method does outperform previous active learning schemes. It is also clear why it is performing better (better acquisition in the overlap region).

**Essential References Not Discussed:**

N/A

**Experimental Designs Or Analyses:**

The experiments seem sound.

**Methods And Evaluation Criteria:**

The benchmarks are standard for these tasks and PEHE is a standard metric for ITE.

**Other Comments Or Suggestions:**

N/A

**Other Strengths And Weaknesses:**

N/A

**Questions For Authors:**

- What is Gamma in L228? I don't believe this is defined anywhere.
- I don't really understand what section 4.2 is trying to solve? It is shown that the covering depends on the data distribution in section 4.4 but statements such as "approximating the full coverage given the relatively smaller fixed covering radius" (L238 RHS) does not make sense to me.
- This problem is compounded as I dont believe P(A^t_F) (and other variants) are properly defined, are these the normalised means over the corresponding covering radiuses?

**Relation To Broader Scientific Literature:**

Although the idea of using covering sets for active learning is not new, using it for the counterfactual covering seems to be novel. It is also demonstrated that this leads to lear performance improvements.

**Theoretical Claims:**

The proofs seem correct.

---

> ### Author Rebuttal · Authors · 2025-03-31
>
> **Q1:** $\Gamma$ is a pseudo-operator defined in the "Input'' section of the Algorithm 1, i.e., $\Gamma=\arg\max\min d(\cdot,\cdot)$. We use $\Gamma$ as a shorthand for the distance-based radius defined underneath Eq. (8) (line 203 right column) to keep Algorithm 1 concise. Take line 6 of Algorithm 1 as an example, to reduce radius $\delta_{(1,1)}$, by applying $\Gamma$ as the selection criteria, the selected sample from $\mathcal{D}_ {1} \setminus\mathcal{S}_ {1}$ is $a=\arg\max_{a'\in\mathcal{D}_ {1}\setminus\mathcal{S}_ {1}}\min_{j\in \mathcal{S}_ {1}} d(\mathbf{x}^{t=1}_ {a'},\mathbf{x}^{t=1}_ {j})$, where $D_ {1}$ and $S_ {1}$ are respectively the unlabeled and labeled sets for $t=1$.
>
> **Q2:** As per line 273 (left column), the informative bound in Theorem 3.3 consisting of four covering radii is derived under the full coverage, i.e., all actively acquired samples should together cover 100\% of the full data space (we will mathematically explain the full coverage in Q3). Thus, the straightforward solution in Algorithm 1 strictly maintains 100\% coverage requirement using currently the smallest radii (in Definition 3.1 and 3.2) at every acquisition step, while progressively minimizing the four radii via newly acquired samples. However, due to this full coverage requirement, Algorithm 1 cannot work effectively on real-world datasets where inter-group distributions do not align well (Figure 2 and lines 255-269 left column). Thus, in Section 4.2, we further build Algorithm 2 (FCCM) upon Algorithm 1 as a solution. **Clarifying the statement:** Instead of computing the minimal radii in Algorithm 1, Algorithm 2 fixes the radii by a small constant when performing data acquisition. Though the 100\% coverage requirement cannot be fully satisfied in early acquisition steps, as the acquisition expands, we can effectively increase the coverage of the data space. So as shown in Figure 3(b), FCCM can achieve up to 25\% bound reduction compared to Algorithm 1 with only a negative $1\%$ coverage gap (99\% coverage). Thus, in line 238 (right column) we state ``given the relatively smaller fixed covering radius'', we can '' approximate (approaching) the full coverage''. We will add more explanations to those parts to better motivate Section 4.2 in an updated version.
>
> **Q3:** For each treatment group $t$,$P(\mathcal{A}^{t}_ {F})$ is the proportion of the data points from the unlabeled pool set $\mathcal{D}_ {t}$ that has been covered by the training set $S_{t}$ in the covering ball $\mathcal{A}^{t}_ {F}$, i.e., $P(\mathcal{A}^{t}_ {F})=\frac{|\mathcal{A}^{t}_ {F}|}{|\mathcal{D}_ {t}|}$. Analogously, $P(\mathcal{A}^{t}_ {CF})=\frac{|\mathcal{A}^{t}_ {CF}|}{|\mathcal{D}_ {1-t}|}$, then the mean coverage rate $P(\mathcal{A})=\frac{1}{4}(P(\mathcal{A}^{t=1}_ {F})+P(\mathcal{A}^{t=0}_ {F})+P(\mathcal{A}^{t=1}_ {CF})+P(\mathcal{A}^{t=0}_ {CF}))$. We will add this formal definition to complement our textual descriptions in lines 266-271 (right column) to make it easier to follow. A further note is that, for Algorithm 1, we have the full coverage, i.e., $P(\mathcal{A})=1$, strictly satisfied at every acquisition step, while Algorithm 2 -- FCCM is to maximize the mean coverage rate $P(\mathcal{A})$ to approximate the full coverage ($P(\mathcal{A})=1$) given the four fixed small covering radii to constitute a lower bound value.

---

### Decision · Program_Chairs · 2025-05-01

**Decision:**

Accept (poster)

**Comment:**

This paper proposes an active learning approach tailored to enhance the treatment effect estimation from observational data, when labeling outcomes is costly. The reviewers found the paper to be novel, with both the theoretical analysis and experimental results being sound. Most of the reviewers' concerns were satisfactorily addressed in the rebuttal. However, the authors are encouraged to incorporate the reviewers' suggestions in the final version to enhance the clarity of this paper.